# TABS: Strategic Game-Based Multi-Stage Reinforcement Learning Challenge

## Abstract

The design of environments plays a critical role in shaping the development and evaluation of reinforcement learning (RL) algorithms. While existing benchmarks have supported significant progress in both single-agent and multi-agent settings, many real-world systems involve multi-stage structures with tightly coupled decision points at each stage. These settings require agents not only to perform well within each stage but also to coordinate effectively across them. We introduce the Totally Accelerated Battle Simulator (TABS), a complex multi-stage environment suite implemented in JAX to enable accelerated training and scalable experimentation. Each TABS task consists of sequential stages with interdependencies, where only the output of one stage is forwarded to the next. This multi-stage structure makes effective exploration challenging, often steering agents toward locally optimal behaviors that limit overall performance. Our empirical analysis shows that standard RL baselines struggle to solve TABS tasks, illustrating the difficulty of learning coherent strategies across interdependent stages. TABS provides a controlled and extensible framework for studying multi-stage decision-making challenges, supporting future research on RL methods capable of operating effectively in structured domains. Our code is available at: https://anonymous.4open.science/r/TABS-0E4B.

## 1 Introduction

Deep reinforcement learning (RL) provides a scalable framework for solving complex sequential decision-making problems by leveraging reward functions to specify desired behaviors. While prior work has achieved impressive results in single-agent settings (Mnih et al., 2013; Schulman et al., 2017; Hessel et al., 2018; Hafner et al., 2025), subsequent research has increasingly focused on multi-agent scenarios (Rashid et al., 2020; Yu et al., 2022; Gallici et al., 2024), reflecting the prevalence of environments where multiple agents must interact, cooperate, or compete. To support progress in this direction, multi-agent game environments have been developed (Samvelyan et al., 2019; Carroll et al., 2019; Kurach et al., 2020; Bard et al., 2020; Ellis et al., 2023), introducing a variety of challenges for RL agents, including partial observability, long-horizon decision-making, high exploration requirements, and the need for effective coordination. These environments have driven significant advancements in algorithmic research, and as a result, many of these challenges have been extensively studied (Kuba et al., 2021; Yu et al., 2022).

Despite notable progress in addressing such challenges, existing environments typically address these issues in isolation. Crucially, real-world systems are often highly complex and correlated across multiple stages, emphasizing the need for methods capable of generalizing across this spectrum. For example, playing sports such as soccer, baseball, and hockey consists of several stages: training, squad selection, strategy building, and gameplay. The objective is to progress through these stages sequentially and ultimately secure victory in the game. The overall success of the process depends not only on the quality of individual contributions but also on how effectively these contributions are aligned and integrated.

This sequential interdependence introduces multi-stage decision-making challenges, requiring agents to consider their long-term consequences across stages. Such multi-stage structures pose challenges not only in designing agents that can operate across heterogeneous task domains, but also in handling cross-stage dependencies that influence downstream performance and demand ef-

ficient exploration strategies. These complexities highlight the need for environments and methods that can effectively model and learn in settings with tightly coupled, stage-wise dynamics.

We introduce *Totally Accelerated Battle Simulator (TABS)*, inspired by the popular strategic simulation game (Landfall Games, 2021), as a highly complex environment specifically designed to present novel and challenging tasks for RL agents. TABS is structured into three sequential stages: *TABSUnitComb*, where agents select unit compositions under a given budget; *TABSUnitDeploy*, where the chosen units are spatially arranged on the battlefield; and *TABSBattleSimulator*, where agents control units in real-time combat. TABS poses three key challenges: **(1) Sequentially interdependent processes**, where decisions made in earlier stages directly constrain and shape the strategic possibilities of subsequent stages; **(2) Entangled credit assignment**, where the environment is structured as a multi-stage process and the final return is determined only after all stages have been completed. This makes it difficult to design training methods that accurately attribute feedback to the decisions made at each individual stage. and **(3) Exploration under multi-stage structure**, where exploration becomes less intuitive and more difficult as the agent must operate across stages that are sequentially organized and interdependent. This poses substantial obstacles to the discovery of globally effective strategies. Overall, TABS instigates not only the challenges present in previous benchmarks but also those arising from interconnected multi-stage structures, further complicating the design of agents. To analyze these challenges in practice, we consider two straightforward training methodologies: simultaneous training, in which all stages are updated jointly, and alternating training, in which each stage is optimized sequentially while the others are held fixed.

Complex environments with long horizons and sequential interdependencies incur substantial computational overhead, limiting the scalability of training and evaluation. In particular, multi-stage environments with tightly coupled stages and stage-specific transitions tend to slow down simulation, as their structure often prevents efficient batching and parallelization. To address this, we implement our environments in JAX (Bradbury et al., 2018), enabling efficient execution on GPUs. This end-to-end JAX-based pipeline substantially accelerates experimentation while maintaining scalability across diverse settings.

To summarize, this paper summarizes the following contributions:

- **A Novel and Challenging Multi-Stage Environment**: We introduce *TABS*, a complex multi-stage environment that poses novel challenges by integrating strongly interdependent stages, which result in entangled credit assignment and highly demanding exploration.
- **Role-appropriate Heuristic Policy**: We provide a heuristic policy for the multi-agent simulation stage that increases the strategic complexity of opponent behaviors, requiring agents to adapt their tactics accordingly.
- **Evaluation Across a Set of Baselines**: We thoroughly benchmark the performance of a representative selection of baselines on our environment, providing insights into learning in highly interdependent multi-stage settings where both observation and action spaces vary substantially between stages.
- **High-Speed Computation via JAX Implementation**: We implement our open-source environment and baselines using the JAX framework, enabling efficient computation on hardware accelerators such as GPUs.

## 2 BACKGROUND

**Strategic Games**  A strategy game is a genre in which players' uncoerced and autonomous decision-making skills play a central role in determining outcomes. Success often depends on the ability to balance short-term tactical actions with long-term strategic objectives, while simultaneously anticipating and responding to the behavior of opponents. In reinforcement learning (RL) research, strategic games have become widely used environments for evaluating agents' decision-making capabilities in complex and uncertain environments. These games typically exhibit high-dimensional state and action spaces, stochastic dynamics, imperfect information, and multi-agent interactions, closely reflecting many of the challenges present in real-world systems.

**JAX-based Environments**  JAX (Bradbury et al., 2018) is a Python library for high-performance numerical computing and large-scale machine learning, providing accelerator-based array computation and program transformation. It compiles Python code into Accelerated Linear Algebra (XLA),

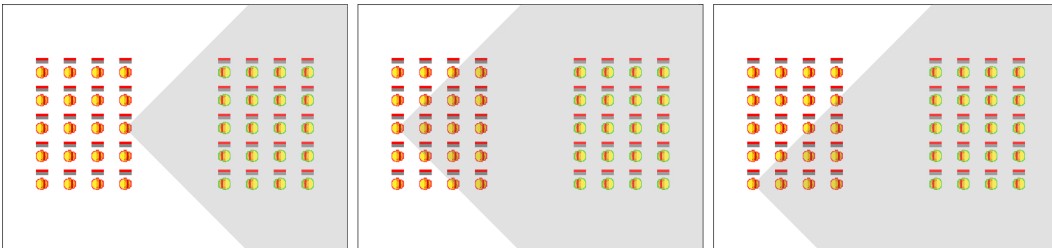

Figure 1: Visualization of a unit's field of view under different initial deployment locations. The agent has only partial observability, restricted to units within its field of view. The fan-shaped region illustrates how initial observations are constrained by deployment placement.

allowing efficient execution on hardware accelerators such as GPUs and TPUs. RL research, the runtime of simulations and algorithms is critical, as it directly affects the efficiency, scale, and feasibility of experiments. RL training typically requires extensive environment interactions, and long, computationally intensive runs can impede research progress. JAX-based environments (Bonnet et al.; Matthews et al., 2024; Rutherford et al., 2024) mitigate these challenges by enabling the entire RL pipeline to run on GPUs, supporting massive parallelization of trajectory collection, eliminating GPU–CPU transfer bottlenecks, and leveraging just-in-time (JIT) compilation throughout the training process. A **detailed discussion of related work** is provided in Appendix B.

## 3 TOTALLY ACCELERATED BATTLE SIMULATOR

Our proposed environment is composed of three sequentially connected stages, TABSUnitComb, TABSUnitDeploy, and TABSBattleSimulator, which together embody the challenges discussed in Section 1. Each stage is defined by its own observation space, action space, and transition dynamics; however, the reward signal is provided only after the entire pipeline has been executed. These environments are deliberately designed to be strongly interdependent, thereby creating tasks that necessitate methods capable of modeling about interdependencies across stages. In Appendix A, we formally describe the decision process.

### 3.1 UNIT COMBINATION AND DEPLOYMENT

The first stage, TABSUnitComb, requires the agent to construct an allied composition by selecting units under a given budget. At each decision step, the agent chooses which unit to purchase based on the remaining budget, the current list of selected units, the specifications of all units, and the fully observed enemy composition. This process continues until the budget is exhausted or no further purchases are possible. The resulting unit composition serves as the initial configuration for all downstream stages.

The second stage, TABSUnitDeploy, tasks the agent with strategically placing the selected units on the ally battlefield. At each step, the agent determines a deployment position for the next unit, conditioned on the remaining units to place, the current positions of already deployed allies, the fully observed enemy layout, and the specifications of all units. Deployment is finalized once all units have been placed. A unique aspect of this stage is spatial influence of deployment position: each unit's field of view in TABSBattleSimulator is a directional, fan-shaped region, oriented along its facing direction. As a consequence, the initial observations at the onset of the battle exhibit high sensitivity to the deployment configuration, as

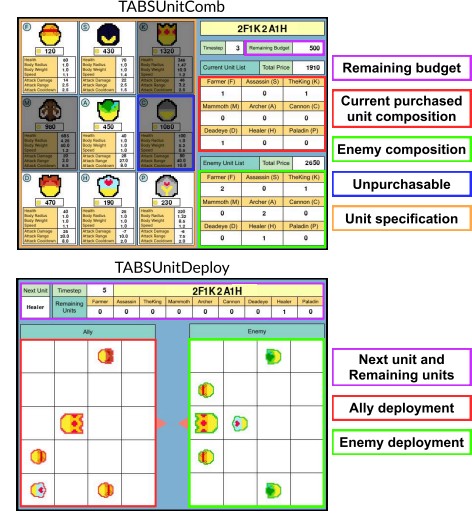

Figure 2: Visualization of TABSUnitComb and TABSUnitDeploy. The full size image is in Appendix D.

illustrated in Figure 1. Additional visualizations of initial observations under various deployment configurations are provided in Appendix A.1.

## 3.2 MULTI-AGENT BATTLE SIMULATION

The final stage, TABSBattleSimulator, requires multiple agents to engage in battle against enemy forces, initialized with the unit composition and spatial deployment determined in the preceding stages. During the battle, allied agents interact with enemies in a decentralized manner. This stage is characterized by high-dimensional shaped partial observations, discrete–continuous hybrid action spaces, and complex interaction systems. Each agent possesses a partial, fan-shaped observation field oriented along its facing direction, analogous to a first-person perspective, illustrated as the gray region in Figure 1. Within this field, agents observe the current status (e.g., remaining health points) and specifications of all visible allies and enemies. As a result, agents must actively rotate and navigate the environment to detect and engage enemy units effectively.

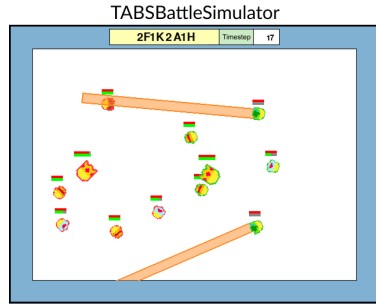

Figure 3: Visualization of TABS-BattleSimulator. The full size image is in Appendix D.

TABSBattleSimulator defines a hybrid action space comprising six discrete actions (four directional movements, attack or heal, and idle) coupled with a continuous rotation action. The movement primitives correspond to fixed step displacements without acceleration dynamics, ensuring consistent translational motion across actions. The rotation control is parameterized as a continuous variable constrained within $[-\pi/12, \pi/12]$, allowing incremental orientation adjustments at each timestep. Discrete and continuous actions can be executed concurrently, with the discrete action applied first followed by the rotation update. Such hybrid action spaces are prevalent in real-world domains (Li et al., 2021), including applications in games (Masson et al., 2016; Xiong et al., 2018), as they enable agents to simultaneously perform discrete behaviors while fine-grained directional adjustments in continuous space.

An interesting aspect of this stage is its interaction system among units, which incorporates a non-targeted attack and healing mechanism, as well as a pushing mechanism based on each unit's body mass. While many existing battle simulation environments rely on explicit targeting systems (Berner et al., 2019; Ellis et al., 2023; Rutherford et al., 2024), in TABS each unit attacks or heals a target only when provided the target is located within its rectangular attack or healing field and its cooldown has elapsed. To successfully execute valid attacks, agents must coordinate their movements, including rotations, with precise attack timing. Agents can exploit their embodied mass to impede the movement of opponents, while incapacitated units remain within the arena as dynamic obstacles that can be displaced through contact forces. Agents receive a shared reward proportional to the difference between the total health ratios of allies and enemies, incentivizing successful attacks or heals. At the end of an episode—when either all allies or all enemies are incapacitated—agents receive an additional binary reward reflecting the win–loss outcome.

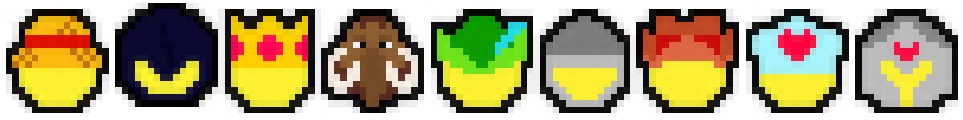

Figure 4: Unit portraits with names and abbreviations from left to right: Farmer (F), Assassin (S), TheKing (K), Mammoth (M), Archer (A), Cannon (C), Deadeye (D), Healer (H), and Paladin (P).

**Unit types** TABS provides a total of nine distinct units, comprising four melee units, three ranged units, and two supporter units. Each unit is characterized by multiple specification components, encompassing physical attributes as well as attack or healing capabilities. We design unit specifications such that, broadly, melee units possess high health and speed to effectively close distance and engage targets, whereas ranged units have lower health and speed but benefit from extended attack range. Each unit is assigned a price proportional to its potential effectiveness. Agents must account

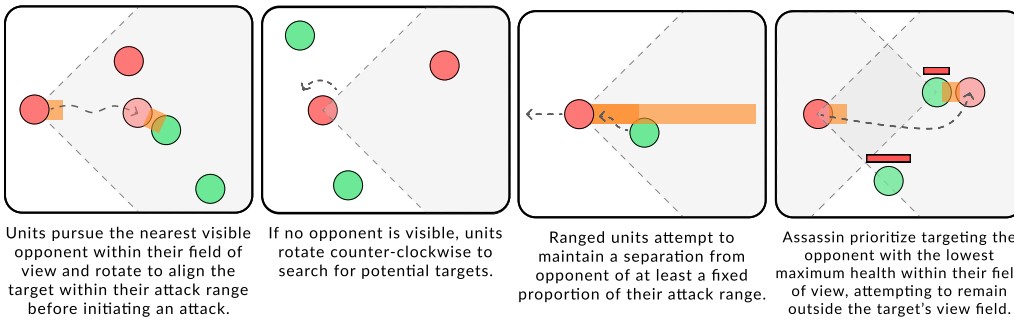

Units pursue the nearest visible opponent within their field of view and rotate to align the target within their attack range before initiating an attack.

If no opponent is visible, units rotate counter-clockwise to search for potential targets.

Ranged units attempt to maintain a separation from opponent of at least a fixed proportion of their attack range.

Assassin prioritize targeting the opponent with the lowest maximum health within their field of view, attempting to remain outside the target's view field.

(a) Standard behaviors

(b) Special behaviors for ranged units and Assassin

Figure 5: Operation of the TABS heuristic policy. Units of the same color belong to the same team. The gray region indicates a unit's field of view, and the orange region denotes its attack range.

for both unit attributes and prices when selecting compositions, balancing cost-effectiveness with strategic potential. Detailed specifications of each unit are provided in Appendix A.2.

### 3.3 ROLE-APPROPRIATE HEURISTIC POLICY

In the TABSBattleSimulator, we provide a role-appropriate heuristic policy. This policy can be applied in multiple ways: assigning it to the opposing side simplifies the problem into a cooperative multi-agent setting, while assigning it to all units reduces TABS to a purely discrete-action problem (TABSUnitComb and TABSUnitDeploy). To ensure diversity and sophistication in their strategies, the policy is differentiated across unit types, with behaviors tailored to roles defined by each unit's specifications. The StarCraft Multi-Agent Challenge (SMAC) (Samvelyan et al., 2019), which is similar to the TABSBattleSimulator, is a widely adopted benchmark for evaluating cooperative behaviors in multi-agent combat environments. However, Rutherford et al. (2024) (SMAX) pointed out limitations of the original benchmark, including the failure of agents to actively pursue enemy units, which led to unrealistic combat dynamics (Samvelyan et al., 2019), and the unrealistic assumption of access to global observations, thereby encouraging more aggressive and realistic behaviors (Ellis et al., 2023). Although SMAX mitigates these limitations, it still applies uniform decision rules across all unit types, without accounting for their distinct roles or capabilities. As a result, it fails to fully exploit the strategic potential of heterogeneous unit compositions, limiting both the expressiveness and complexity of scenario design.

Our heuristic policy incorporates a heterogeneous set of units with distinct functional roles—such as melee, ranged, and supporter units—each necessitating role-specific decision-making strategies. The integration of expert-designed, role-aware heuristic policies in TABS enables the environment to support richer and more strategically demanding scenarios. These heuristics induce opponent behaviors that differ meaningfully across unit roles, thereby fostering task designs in which unit selection, deployment, and battle decisions are interdependent and contingent on the configuration of opposing forces. This functionality renders TABS a flexible and challenging testbed for evaluating multi-stage decision-making and multi-agent coordination.

Figure 5 explains the operation of our role-appropriate strategic heuristic policy. Units are broadly categorized into two types: melee and ranged. We categorize these two types based on whether their attack or heal range exceeds a predefined threshold. Both types follow a set of standard behavioral rules, as shown in Figure 5a: they pursue the nearest visible opponent (or injured ally in the case of supporters) and rotate to align the target within their attack or heal range, thereby making it attackable or healable. If no target is visible, they rotate counter-clockwise to search for potential targets. Units always execute an attack action immediately once the cooldown period has elapsed, provided the target is within attack or heal range; otherwise, they perform either a movement and rotation action. Support units exclusively target allies, restoring their health upon a successful action. If at least one ally is injured, supporters restrict their focus to those units, prioritizing the nearest injured ally for healing. We provide both melee and ranged supporter types, each adhering to the strategic behavioral rules of their respective unit category.

Table 1: TABS scenarios and their corresponding unit compositions. The numbers denote the quantity of each unit, while the letters represent unit types: Farmer (F), Assassin (S), The King (K), Mammoth (M), Archer (A), Cannon (C), Deadeye (D), Healer (H), and Paladin (P).

| Scenario | Composition Design |
|---|---|
| 2F1K2A1H | Classic composition (tanker, dealer, supporter) |
| 1K2S | Long-range dealer attack composition |
| 1M2C1P | Frontline tank and rear high-attack ranged attackers |
| 7F2D1H | War of attrition |

To enable units to exhibit advanced, strategy-driven behaviors, we introduce two specialized strategies, as illustrated in Figure 5b. Ranged attackers retreat to maintain a distance from opponents of at least a fixed proportion of their attack range. This "kiting" behavior forces opponents to execute more complex movements to engage effectively. The Assassin, a melee unit, follows a distinctive strategy consistent with its name: it targets the rear of the visible opponent with the lowest maximum health, leveraging its high movement speed to execute opportunistic attacks while attempting to remain outside the target's field of view. This behavior introduces additional challenges, requiring careful and deliberate unit composition and deployment strategies. We inject controllable noise via hyperparameters, thereby introducing stochasticity into the behavior and modulating the behavioral optimality of the heuristic policy. Further implementation details of the heuristic policy are provided in Appendix A.3.

### 3.4 SCENARIOS

We provide a set of predefined scenarios that take into account the attributes and strategic behaviors of each unit type. These scenarios are categorized based on the available budget relative to the total cost of the enemy composition. The enemy composition and deployment in each scenario are carefully crafted to achieve specific objectives. Table 1 presents the predefined scenarios along with the underlying design intentions. Their designs are inspired by strategies that are commonly employed in real strategic games. Each scenario is associated with three different budget levels reflecting varying levels of budget (abundant, medium, tight). The medium budget level is set equal to the total cost of the enemy composition, enabling performance comparison under same conditions. The abundant budget level provides additional resources, allowing the purchase of more expensive units if cheaper enemy units are removed, while the tight budget level imposes the opposite constraint.

A higher budget allows the agent to recruit expensive units and field larger forces, thereby reducing the difficulty of configuring an effective ally troop. Conversely, a lower budget constrains feasible unit compositions, forcing the agent to carefully balance cost-effectiveness with strategic potential, while also mastering deployment and combat against a numerically or qualitatively superior enemy force. In TABS, the entire process is initialized according to the specified budget characteristics. In Appendix A.4, we provide detailed descriptions of each scenario along with simulation results obtained using manually crafted ally compositions and deployments, evaluated under varying budget constraints.

## 4 EXPERIMENTS

We conduct a series of experiments to evaluate the performance of baseline algorithms in our proposed environment. First, we discuss the challenges of agent design and training approaches in the multi-stage environment. Second, we compare training results across scenarios under varying budget levels. Third, we analyze the exploration challenges induced by our environment's design. Finally, we conduct a scalability study to demonstrate that the environment can efficiently leverage parallel execution.

### 4.1 TRAINING METHODS

TABS, a multi-stage environment, presents several inherent challenges for training agents. The initial observations of later environments depend on the outcomes of preceding stages, further com-

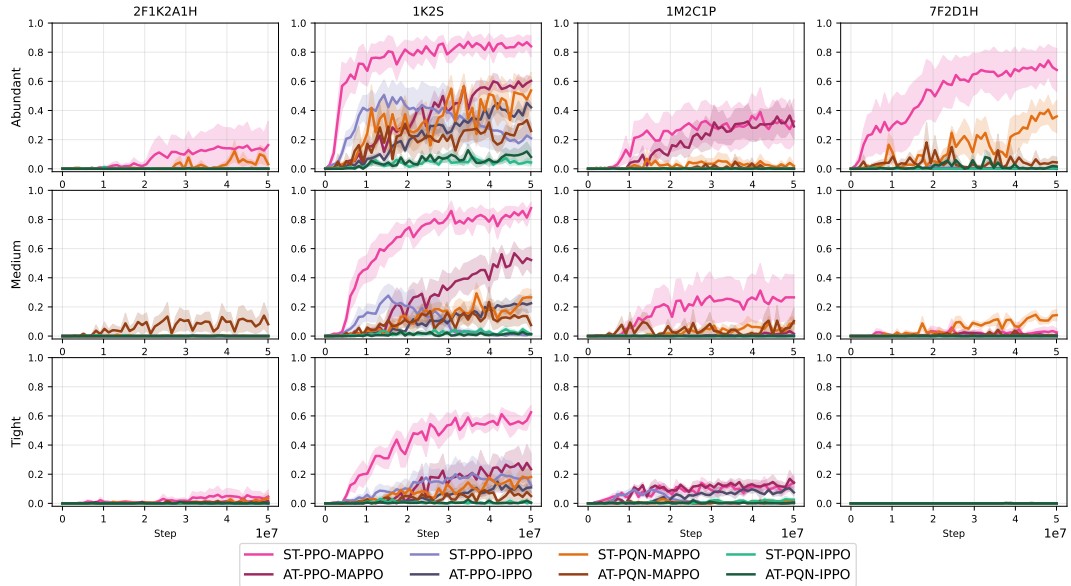

Figure 6: Average episode win rates of baselines trained with alternating and simultaneous methods across four scenarios and three budget levels. Results are averaged over five random seeds, with shaded regions indicating the standard error.

pounding the training difficulty. Specifically, TABSUnitDeploy operates on the output of TABSUnitComb (the selected ally unit composition), and TABSBattleSimulator operates on the output of TABSUnitDeploy (the deployed ally units). Thus, the initial state of each stage varies according to decisions made in preceding stages, which reduces training stability. Training a single policy across all stages is challenging, because each stage in TABS is defined by distinct state and action spaces as well as heterogeneous task settings (ranging from single-agent to multi-agent control). This stands in contrast to prior benchmarks such as Cobbe et al. (2020), which focus on procedurally generated game-like environments that maintain consistent state and action representations across tasks.

A straightforward approach to handle this is to employ a stage-conditioned mixture of heterogeneous stage policies. We first collect trajectories by rolling out the entire pipeline and assign the win–lose outcome to the last state–action pair of experiences in TABSUnitComb and TABSUnitDeploy, while all other pairs receive a reward of 0. This design implies that feedback for early-stage decisions is inherently delayed and aggregated, since the final outcome is determined only after the entire multi-stage pipeline has been completed. Moreover, because each stage involves distinct policy structures and decision spaces, devising efficient exploration strategies becomes particularly challenging.

In our experiments, we investigate two straightforward approaches for training the agent, formulated as a stage-conditioned mixture of stage policies: simultaneous training and alternating training. In simultaneous training, all policies are updated jointly using their respective trajectories from the same episode. In alternating training, by contrast, each policy is updated exclusively for several consecutive iterations while the others remain frozen; once updates for one policy are completed, training shifts to the next in an alternating cycle. We further adopt controlled experimental settings to facilitate a more precise analysis of the challenges posed by our environment. In particular, we employ fixed scenarios and hold the opponent's policy constant by using a heuristic controller.

**Baselines** We evaluate a suite of standard baselines across the stages. For TABSUnitComb and TABSUnitDeploy, we employ PPO (Schulman et al., 2017) and PQN (Gallici et al., 2024), while for TABSBattleSimulator we adopt MAPPO (Yu et al., 2022) and IPPO (De Witt et al., 2020). For end-to-end training, we construct composite agents by combining the corresponding baselines across all stages and train the resulting set of stage policies using both simultaneous and alternating training methods. Since we employ the same algorithms for both TABSUnitComb and TABSUnitDeploy, we obtain a total of eight baselines from the combination of four algorithms with two training methods. We denote each baseline by its training method—ST (simultaneous training) or AT (alternating training)—followed by the adopted policies (e.g., ST-PPO-MAPPO).

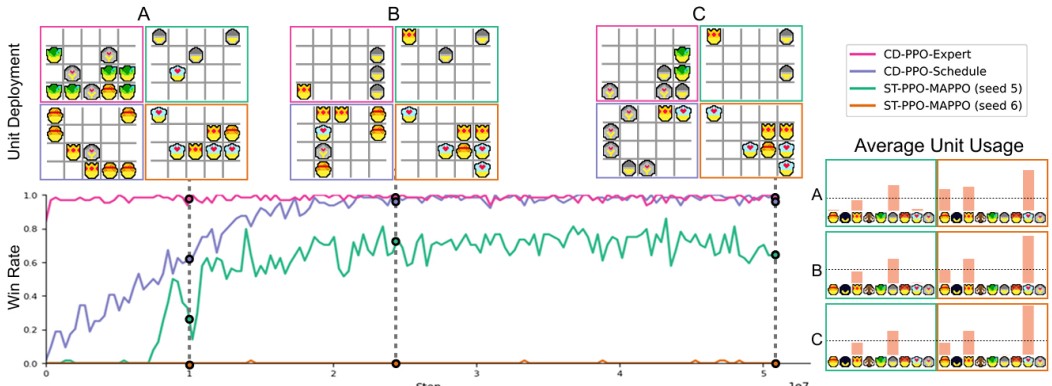

Figure 7: Illustration of training outcomes across different random seeds in `1M2C1P` scenario with the abundant budget level. Each panel shows unit deployment (top), the corresponding win rate (bottom), and average unit usage (right).

## 4.2 TRAINING IN MULTI-STAGE ENVIRONMENT

We evaluate the baselines across scenarios with varying budget levels, using two training methods. As shown in Figure 6, ST-PPO-MAPPO achieves the strongest performance across most scenarios, except in `2F1K2A1H` and `7F2D1H` with medium budget levels, where PQN-MAPPO performs better. Overall, PPO generally outperforms PQN, although PQN occasionally succeeds in environments where PPO fails. Comparing MAPPO and IPPO, MAPPO demonstrates superior performance. Regarding training approaches, simultaneous training typically leads to more efficient learning than alternating training and exhibits lower performance variance. We further provide several interesting metrics including not only episode return but also first-kill rate, total episode damage dealt, attack success rate, and average unit counts for each unit. We illustrate these results in Appendix C.6.

Baselines achieve higher performance in `1K2S` compared to other scenarios. The `1K2S` consists of three melee units without any support units, making it relatively easier to learn effective strategies against them than against ranged attackers. As expected, performance improves with larger budgets, and under abundant budget conditions, agents can easily defeat their opponents. In particular, ST-PPO-MAPPO achieves over a 60% win rate in `7F2D1H-Abundant`, where `7F2D1H` represents a large-scale troop scenario featuring the maximum number of enemy units among all settings.

Figure 6 presents that most baselines exhibit high variance across random seeds. We observe that simple entropy regularization did not alleviate this phenomenon, implying the inherent difficulty in exploration. We attribute this high variance to the multi-stage nature of the environment, where the output of one stage becomes the constrained input for the next. This dependency forces agents to specialize to the narrow distributions induced by earlier stages, thereby limiting generalization and leading to convergence toward local optima. While such phenomena can also arise in non-staged environments, the inherent pipeline structure of multi-stage settings makes effective exploration particularly challenging.

## 4.3 EXPLORATION IN MULTI-STAGE ENVIRONMENT

To analyze the exploration difficulty in TABS, we conduct additional experiments under controlled settings. Specifically, we train agents in TABSUnitComb and TABSUnitDeploy while employing the heuristic policy with varying levels of stochasticity: constant low noise (Expert) and linearly decreasing noise up to the Expert level by approximately 50% of training (Schedule). CD-PPO denotes training PPO in the first two stages while employing the heuristic policy with varying levels of stochasticity. Figure 7 presents unit deployment and average unit usage as a function of win rate at specific environment steps across baselines in `1M2C1P` under the abundant budget level. This scenario features a balanced unit composition consisting of two melee units (a tanker and a support) and two ranged attackers.

The results in TABSUnitComb significantly influence the configuration of subsequent stages and constrain the available search space, highlighting the importance of exploration in this stage. Al-

though the agent attempts to explore diverse unit combinations during early training, it tends in TABS to repeatedly commit to combinations that yield higher initial returns due to increasingly stable initializations, thereby failing to acquire proficiency across a broader set of units. While CD-PPO purchases a diverse set of unit types and, despite poor early performance, CD-PPO-Schedule successfully converges to an optimal policy as the proficiency of the heuristic policy increases linearly, ST-PPO-MAPPO increasingly relies on a limited subset of units as training progresses, resulting in a lower win rate, as shown in Figure 7.

To analyze seed sensitivity, we select two seeds that exhibit large performance discrepancies and compare their final average unit usage at three checkpoints. We observe that both seeds struggle to explore diverse unit compositions and deployment strategies. Early-stage restrictions imposed by the chosen seeds lead to limited initial compositions, resulting in significant performance discrepancies. From the multi-stage perspective, early biases arising from the structural difficulty of exploration are reinforced, preventing the agent from revisiting alternative strategies. Since the stages are distinct and sequentially coupled, designing direct exploration strategies is inherently challenging, highlighting the need for more efficient mechanisms specifically tailored to TABS.

### 4.4 Environment Parallelization Experiments

To assess the scalability of our environment, we conduct experiments measuring throughput as a function of the number of parallel environments executed on a single GPU. Specifically, we report the effective environment steps per second while varying the degree of parallelization. Three stages are denoted as Comb, Deploy, and Battle, respectively, and $n$ represents the maximum number of units in the final stage. The results, summarized in Figure 8, show that throughput steadily increases with the number of parallel environments, though the rate of improvement diminishes once hardware constraints such as GPU memory bandwidth become a limiting factor. These findings confirm that our environment can fully leverage accelerators, enabling efficient large-scale training of agents. Details of the hardware configuration are provided in Appendix C.

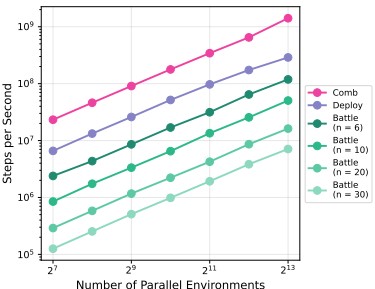

Figure 8: Speed under environment parallelization on an RTX 4090.

## 5 Conclusion

We introduce TABS, a multi-stage reinforcement learning environment that explicitly captures sequential interdependencies across distinct stages, entangled credit assignment, and provides a role-appropriate heuristic policy that serves as a competitive control mechanism. Our experiments demonstrate that training in such settings entails inherent challenges, as agents can easily converge to suboptimal strategies due to the difficulty of exploration in strongly interdependent multi-stage environments. These results underscore the need for algorithms capable of both efficient exploration and robust learning in multi-stage settings. In addition, we show that the environment scales effectively under parallelization, enabling efficient and accelerated large-scale training on GPUs. We hope that TABS will facilitate research in areas including end-to-end training, entangled credit assignment, handling heterogeneous action and observation spaces, and exploration. We believe that an agent capable of tackling TABS would represent a significant advancement in the field, and we look forward to seeing how the community leverages this benchmark for future developments.

**Future Work**   While the current environment provides functional implementations, opportunities remain for further enhancing the degree of interdependency across stages. As future work, we plan to extend the deployment stage by introducing terrain obstacles, requiring agents to account for spatial constraints during unit placement, and to enrich the battle stage by allowing agents to strategically exploit these obstacles in combat. In addition, we aim to enhance the environment engine to more closely align with Landfall Games (2021), for example by incorporating unit-specific skills beyond static attributes and restricting agents to a first-person point of view. These extensions are expected to amplify cross-stage dependencies, thereby intensifying the inherent challenges of multi-stage decision-making.

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

# A  DETAILS ON TABS

We focus on sequentially combined environments inspired by strategy-based games that naturally incorporate interdependent substages. A prominent example is Landfall Games (2021), a popular strategic simulation game released in 2021. Landfall Games (2021) consists of two primary stages: first, players select their own army under a given budget and deploy their units, and second, the game simulates a battle between the deployed units and enemy forces. Decisions made during the early stage propagate forward, creating dependencies that significantly influence subsequent outcomes and require players to balance immediate choices against long-term objectives. These games thus provide natural testbeds for evaluating agents' ability to strategically plan, reason, and generalize across multiple interconnected stages, capturing the complexity and sequential interdependence.

We model the full decision-making process as a Decentralized Partially Observable Markov Decision Process (Dec-POMDP). We first define

$$\mathcal{M}_{\text{comb}} = \langle \mathcal{S}_{\text{comb}}, \mathcal{A}_{\text{comb}}, P_{\text{comb}}, R_{\text{comb}}, \gamma \rangle, \quad \mathcal{M}_{\text{deploy}} = \langle \mathcal{S}_{\text{deploy}}, \mathcal{A}_{\text{deploy}}, P_{\text{deploy}}, R_{\text{deploy}}, \gamma \rangle$$

as single-agent MDPs for TABSUnitComb and TABSUnitDeploy, and

$$\mathcal{M}_{\text{battle}} = \langle \mathcal{S}_{\text{battle}}, \{\mathcal{A}_{\text{battle}}^{(i)}\}_{i=1}^n, O_{\text{battle}}, P_{\text{battle}}, R_{\text{battle}}, \gamma \rangle$$

as a cooperative multi-agent Dec-POMDP for TABSBattleSimulator. We then model the full decision-making process as a Dec-POMDP

$$\mathcal{M} = \langle \mathcal{S}, \mathcal{A}, \mathcal{O}, P, R, \gamma \rangle,$$

with the following components (stage unions and stage-gated transitions):

**State space.**

$$\mathcal{S} = \mathcal{S}_{\text{comb}} \cup \mathcal{S}_{\text{deploy}} \cup \mathcal{S}_{\text{battle}},$$

**Action space.**

$$\mathcal{A} = \mathcal{A}_{\text{comb}} \cup \mathcal{A}_{\text{deploy}} \cup \left( \prod_{i=1}^n \mathcal{A}_{\text{battle}}^{(i)} \right).$$

**Observation space.**

$$\mathcal{O} = \mathcal{S}_{\text{comb}} \cup \mathcal{S}_{\text{deploy}} \cup \mathcal{O}_{\text{battle}},$$

i.e., observations equal states in Comb/Deploy (fully observed), and follow $O_{\text{battle}}$ in Battle. For $s \in \mathcal{S}$,

$$O(o \mid s, a) = \begin{cases} s & s \in \mathcal{S}_{\text{comb}} \cup \mathcal{S}_{\text{deploy}} \\ O_{\text{battle}}(o_{\text{battle}} \mid s, a) & s \in \mathcal{S}_{\text{battle}} \end{cases}$$

**Reward function.**

$$R(s, a) = \begin{cases} R_{\text{comb}}(s, a) & s \in \mathcal{S}_{\text{comb}} \\ R_{\text{deploy}}(s, a) & s \in \mathcal{S}_{\text{deploy}} \\ R_{\text{battle}}(s, a) & s \in \mathcal{S}_{\text{battle}} \end{cases}$$

**Transition Dynamics.** For non-terminal states

$$P(s' \mid s, a) = \begin{cases} P_{\text{comb}}(s'|s, a) & s \in \mathcal{S}_{\text{comb}} \\ P_{\text{deploy}}(s'|s, a) & s \in \mathcal{S}_{\text{deploy}} \\ P_{\text{battle}}(s'|s, a) & s \in \mathcal{S}_{\text{battle}} \end{cases}$$

For terminal states, we define a deterministic stage-transition function $\Phi : \mathcal{S} \times \mathcal{A} \to \mathcal{S}$ such that

$$P(s' \mid s, a) = \begin{cases} \delta\big(s' = \Phi_{\text{comb} \to \text{deploy}}(s, a)\big), & s \in \mathcal{S}_{\text{comb}}, \text{Done}_{\text{comb}}(s), \\ \delta\big(s' = \Phi_{\text{deploy} \to \text{battle}}(s, a)\big), & s \in \mathcal{S}_{\text{deploy}}, \text{Done}_{\text{deploy}}(s) \end{cases}$$

where $\delta(\cdot)$ denotes an indicator distribution.

The objective of the policy $\pi$ is to maximize the expected return $\mathbb{E}[\sum_i^\infty \gamma^i r_{t+i}]$. The policy is defined as a stage-conditioned mixture of sub-policies:

$$\pi(\cdot \mid o) = \mathbf{1}_{S_{\text{Comb}}} \pi_{\text{C}}(\cdot \mid o) + \mathbf{1}_{S_{\text{Deploy}}} \pi_{\text{D}}(\cdot \mid o) + \mathbf{1}_{S_{\text{Battle}}} \pi_{\text{B}}(\cdot \mid o)$$

where $\pi_{\text{C}}, \pi_{\text{D}}, \pi_{\text{B}}$ denote the policies specialized for three stages, respectively.

A.1   INITIAL OBSERVATION IN TABSBATTLESIMULATOR

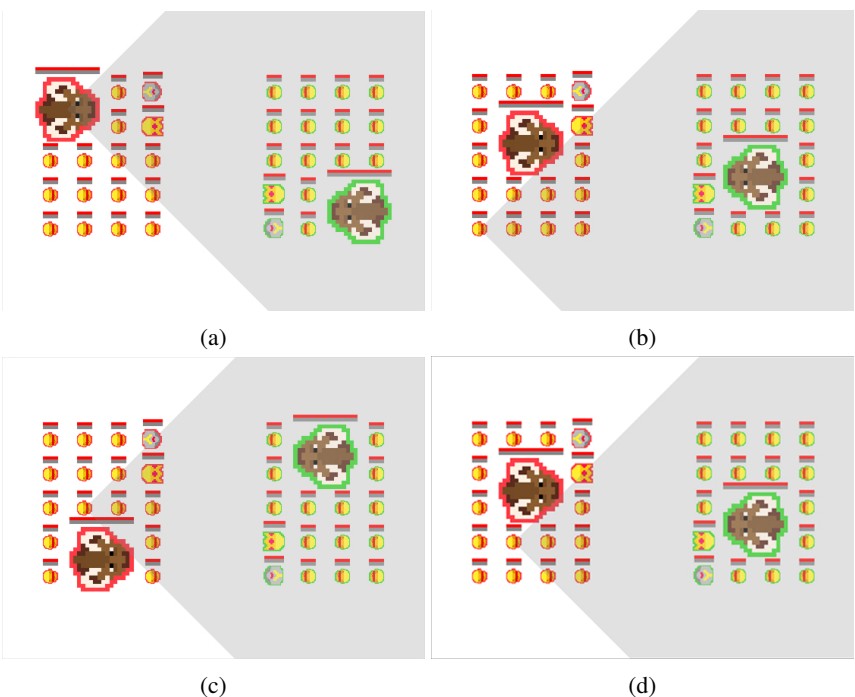

(a)                 (b)

(c)                 (d)

Figure 9: Visualization of unit's field of view under different initial deployment locations and unit types. The fan-shaped region illustrates how initial observations are constrained by deployment placement.

We visualize initial observations under different deployment configurations in Figure 9. Depending on the deployment position, the large unit may or may not fall within the agent's observable field. For instance, the Mammoth is visible within the unit's sight in Figure 9b, Figure 9c, and Figure 9d.

A.2   UNIT TYPES AND SPECIFICATIONS

Each unit is characterized by multiple specification components, including price, health, body radius, body weight, speed, attack damage, attack range, attack cooldown, sight angle, and occupied space. We provide nine predefined units: Farmer (F), Assassin (S), TheKing (K), Mammoth (M), Archer (A), Cannon (C), Deadeye (D), Healer (H), and Paladin (P). The Farmer is a basic unit, serving as the weakest melee attacker but also the most cost-effective option. The Assassin is the fastest unit and has the shortest attack cooldown. TheKing is slightly larger than most other units, with high health and very high melee attack damage, though at a significant cost. The Mammoth is the largest and heaviest unit—approximately four times larger and fifty times heavier than the others—while also being slightly faster, making it well-suited for breaking through enemy formations. The Archer, Cannon, and Deadeye are ranged attackers: the Archer has low health but moderate attack power and long range; the Cannon has the highest attack damage and longest range but is the slowest and most expensive; and the Deadeye has the shortest range but is the fastest among ranged units. The Healer and Paladin are support units with a unique ability: when performing the "attack" action, they restore health to allies instead of dealing damage. Detailed unit statistics are provided in Table 2.

A.3   ROLE-APPROPRIATE HEURISTIC POLICY

In this section, we provide additional details of the heuristic policy that could not be fully covered in the main text.

Table 2: Unit statistics used in TABS. Negative attack damage values correspond to healing effects.

| Name | Price | Health | Body Radius | Body Weight | Speed | Attack Damage | Attack Range | Attack Cooldown | Space Occupied |
|---|---|---|---|---|---|---|---|---|---|
| Farmer (F) | 120 | 60 | 1.0 | 1.0 | 1.1 | 14 | 2.5 | 2.5 | 1 |
| Assassin (S) | 430 | 70 | 1.0 | 1.0 | 1.4 | 22 | 2.5 | 1.5 | 1 |
| TheKing (K) | 1320 | 346 | 1.47 | 10.0 | 1.2 | 46 | 3.2 | 2.5 | 1 |
| Mammoth (M) | 980 | 685 | 4.25 | 50.0 | 1.2 | 20 | 3.0 | 6.5 | 4 |
| Archer (A) | 450 | 40 | 1.0 | 1.0 | 1.0 | 28 | 27.0 | 8.0 | 1 |
| Cannon (C) | 1080 | 100 | 1.0 | 5.2 | 0.5 | 80 | 40.0 | 10.0 | 1 |
| Deadeye (D) | 470 | 40 | 1.0 | 1.0 | 1.1 | 25 | 20.0 | 8.0 | 1 |
| Healer (H) | 190 | 25 | 1.0 | 1.0 | 1.0 | -7 | 10.0 | 2.0 | 1 |
| Paladin (P) | 230 | 220 | 1.32 | 8.5 | 1.2 | -6 | 7.5 | 2.0 | 1 |

**Moving Algorithm**   All units navigate toward their designated target position. Movement is determined by comparing the coordinate differences between the unit's current position and the target position. The axis with the largest absolute difference is prioritized, and the unit moves in the direction that reduces this difference, thereby progressing toward the target location in a stepwise manner.

**Target Position.**   By default, each unit selects as its target position the location aligned with the opponent's facing direction, adjusted by the unit's own radius. Two exceptions apply: (i) the Assassin unit instead selects the position behind the target, opposite to the opponent's facing direction, reflecting its opportunistic playstyle; and (ii) the Healer unit targets the center of its ally rather than its front, ensuring that the healing effect is applied reliably.

**Random Noise.**   The heuristic policy incorporates three forms of controllable stochasticity:

1. **Random action noise.** With probability $\epsilon$, an agent executes one of the movement actions chosen uniformly at random, regardless of the underlying policy. For rotation, an additional perturbation is applied by sampling

$$\delta\theta \sim \mathcal{N}\left(0, \tfrac{1}{\pi}^2\right),$$

   where the sampled value is added to the rotation action at each timestep.

2. **Ranged rotation noise.** For ranged units, we introduce an additional Gaussian perturbation with standard deviation $\epsilon_{\text{ranger}}$, applied on top of the intended rotation action:

$$\delta\theta_{\text{ranger}} \sim \mathcal{N}\left(0, \epsilon_{\text{ranger}}^2\right).$$

   This noise is always combined with the executed rotation, ensuring variability in targeting behavior.

3. **Healer-specific rotation noise.** For ranged healer units, we define a separate hyperparameter $\epsilon_{\text{healer}}$ to control the magnitude of rotation noise:

$$\delta\theta_{\text{healer}} \sim \mathcal{N}\left(0, \epsilon_{\text{healer}}^2\right).$$

   This ensures that healing actions remain stochastic in a manner distinct from offensive ranged units.

Together, these noise components prevent the heuristic policy from becoming overly deterministic or optimal, while allowing fine-grained control of randomness via tunable hyperparameters $\epsilon$, $\epsilon_{\text{ranger}}$, and $\epsilon_{\text{healer}}$.

**Kiting Algorithm**   Ranged units attempt to maintain a safe distance from opponents. Formally, let $d(u, v)$ denote the Euclidean distance between a ranged unit $u$ and an opposing unit $v$, and let $R_u$ be the attack range of unit $u$.

For standard ranged attackers, if

$$d(u, v) \leq \alpha_{\text{ranged}} R_u,$$

the unit does not move toward its designated target position. Instead, it moves in the opposite direction of the intended movement vector, thereby retreating to preserve distance while continuing to threaten the opponent within range.

For ranged healers, a separate threshold parameter $\alpha_{\text{healer}}$ is introduced. Specifically, if

$$d(u, v) \leq \alpha_{\text{healer}} R_u,$$

the healer retreats in a similar manner, but with the additional objective of maintaining sufficient spacing from allied units. This behavior ensures that the healer can continue to provide support while reducing the risk of being caught in close combat or obstructing allied positioning.

**Thresholds for Unit Roles**  Unit roles are determined based on predefined attribute thresholds. Specifically, ranged units are defined as those with an attack range greater than or equal to 10, while assassin units are defined as those with a movement speed greater than or equal to $1.4$.

## A.4    SCENARIOS

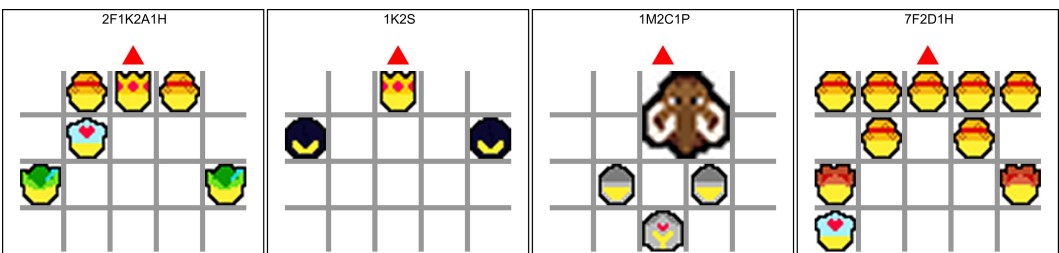

Figure 10: Illustrations of the initial deployments for the four benchmark scenarios. The red triangle denotes the facing direction of the enemy forces.

Table 3: Budget levels (abundant, medium, tight) for each scenario.

| Scenario | Abundant | Medium | Tight |
| --- | --- | --- | --- |
| 2F1K2A1H | 2930 | 2650 | 2320 |
| 1K2S | 2420 | 2180 | 1940 |
| 1M2C1P | 3520 | 3370 | 2570 |
| 7F2D1H | 2450 | 1970 | 1720 |

We provide a set of predefined scenarios that specify the initial unit compositions and deployment layouts for the *enemy* side, which serve as fixed adversaries throughout our experiments. Each scenario determines both the unit types and their spatial arrangement on a fixed grid. The red triangle denotes the facing direction of the enemy forces. Illustrations of the deployments are shown in Figure 10, and the corresponding budget levels for each scenario are summarized in Table 3.

## B    RELATED WORK

**Multi-stage RL Environments**  Deep RL has recently achieved remarkable success across a wide range of complex, long-horizon environments (Machado et al., 2018; Fan et al., 2022). To promote stronger generalization, Cobbe et al. (2020) introduced a suite of procedurally generated game-like environments organized into a sequential pipeline. However, the interconnections between successive stages in their framework are weak, and all stages share a unified action and observation space. In contrast, the stages in TABS exhibit strong interdependencies—where outcomes from earlier stages directly shape later ones—and feature distinct action and observation spaces, posing a significant challenge for agent design.

**Game-based Multi-agent RL Environments**  Game-based environments have become a widely adopted domain for evaluating multi-agent scalability, coordination, cooperation, and collaboration. Prior work has introduced a diverse set of such environments, including StarCraft II (Vinyals et al., 2019), which has inspired multiple environment suites (Samvelyan et al., 2019; Ellis et al., 2023; Rutherford et al., 2024); Overcooked (Ghost Town Games, 2016), widely used for multi-agent coordination research (Carroll et al., 2019; Rutherford et al., 2024); the Google Research Football

environment (Kurach et al., 2020); and traditional card and board games (e.g., poker, tic-tac-toe) (Lanctot et al., 2019). While these works primarily focus on single-stage environments, we target sequentially combined environments, reflecting the interwoven stages characteristic of many real-world problems. Although Xi et al. (2023) investigated a two-stage strategy card game, we extend this setting to a $n$-player environment.

# C EXPERIMENTAL DETAIL

## C.1 ALTERNATING AND SIMULTANEOUS TRAINING

We present pseudocode for both alternating and simultaneous training strategies in Algorithms 1 and 2. In both cases, the policies $\pi_C(\theta_C)$, $\pi_D(\theta_D)$, and $\pi_B(\theta_B)$—corresponding to TABSUnitComb, TABSUnitDeploy, and TABSBattleSimulator stages, respectively—are trained end-to-end through full pipeline rollouts, with the key difference lying in the parameter update schedule.

---

**Algorithm 1** Alternating Training for Multi-Stage Agents

---

**Require:** Agents $\pi_C(\theta_C)$, $\pi_D(\theta_D)$, $\pi_B(\theta_B)$; total outer iterations $T$; update steps per stage: $K_C$, $K_D$, $K_B$.
1: Initialize parameters $\theta_C, \theta_D, \theta_B$
2: **for** $t = 1$ **to** $T$ **do**
3:    // TABSUnitComb training phase
4:    **for** $k = 1$ **to** $K_C$ **do**
5:       Rollout full pipeline $(\pi_C, \pi_D, \pi_B)$
6:       Collect episode trajectories $\mathcal{D}_C$ from TABSUnitComb stage
7:       Set reward of the last episode action as the final TABSBattleSimulator return
8:       Update $\theta_C$ with $\mathcal{D}_C$ (freeze $\theta_D, \theta_B$)
9:    **end for**
10:   // TABSUnitDeploy training phase
11:   **for** $k = 1$ **to** $K_D$ **do**
12:      Rollout full pipeline $(\pi_C, \pi_D, \pi_B)$
13:      Collect episode trajectories $\mathcal{D}_U$ from TABSUnitDeploy stage
14:      Set reward of the last Deploy episode action as the final TABSBattleSimulator return
15:      Update $\theta_U$ with $\mathcal{D}_U$ (freeze $\theta_C, \theta_B$)
16:   **end for**
17:   // TABSBattleSimulator training phase
18:   **for** $k = 1$ **to** $K_B$ **do**
19:      Rollout full pipeline $(\pi_C, \pi_D, \pi_B)$
20:      Collect episode trajectories $\mathcal{D}_B$ from TABSBattleSimulator stage
21:      Update $\theta_B$ with $\mathcal{D}_B$ (freeze $\theta_C, \theta_D$)
22:   **end for**
23: **end for**
24: **return** trained parameters $\theta_C^\star, \theta_D^\star, \theta_B^\star$

---

---

**Algorithm 2** Simultaneous Training for Multi-Stage Agents

---

**Require:** Agents $\pi_C(\theta_C)$, $\pi_D(\theta_D)$, $\pi_B(\theta_B)$; total iterations $T$;.
1:  Initialize parameters $\theta_C, \theta_D, \theta_B$
2:  **for** $t = 1$ **to** $T$ **do**
3:      Rollout full pipeline $(\pi_C, \pi_D, \pi_B)$
4:      Collect episode trajectories $\mathcal{D}_C, \mathcal{D}_D, \mathcal{D}_B$ for each stage
5:      Set reward of the last TABSUnitComb action as the final TABSBattleSimulator return
6:      Set reward of the last TABSUnitDeploy action as the final TABSBattleSimulator return
7:      Update $\theta_C$ with $\mathcal{D}_C$
8:      Update $\theta_D$ with $\mathcal{D}_D$
9:      Update $\theta_B$ with $\mathcal{D}_B$
10: **end for**
11: **return** trained parameters $\theta_C^\star, \theta_D^\star, \theta_B^\star$

---

Table 4: Alternating training common settings

| Setting | Value |
| --- | --- |
| # Parallel environments | 64 |
| # Max ally units | 10 |
| Total iterations $T$ | 100 |
| $K_C$ | 50 |
| $K_D$ | 50 |
| $K_B$ | 50 |
| TABSUnitComb rollout steps | 10 |
| TABSUnitDeploy rollout steps | 10 |
| TABSBattleSimulator rollout steps | 512 |
| Heuristic random action probability $\epsilon$ | 0.1 |
| $\epsilon_{\mathrm{ranged}}$ | 0.5 |
| $\epsilon_{\mathrm{healer}}$ | 0.1 |
| $\alpha_{\mathrm{ranged}}$ | 0.3 |
| $\alpha_{\mathrm{healer}}$ | 0.85 |

Table 5: Simultaneous training common settings

| Setting | Value |
| --- | --- |
| # Parallel environments | 64 |
| # Max ally units | 10 |
| Total iterations $T$ | 1500 |
| TABSUnitComb rollout steps | 10 |
| TABSUnitDeploy rollout steps | 10 |
| TABSBattleSimulator rollout steps | 512 |
| Heuristic random action probability $\epsilon$ | 0.1 |
| $\epsilon_{\mathrm{ranged}}$ | 0.5 |
| $\epsilon_{\mathrm{healer}}$ | 0.1 |
| $\alpha_{\mathrm{ranged}}$ | 0.3 |
| $\alpha_{\mathrm{healer}}$ | 0.85 |

## C.2 POLICY ARCHITECTURE

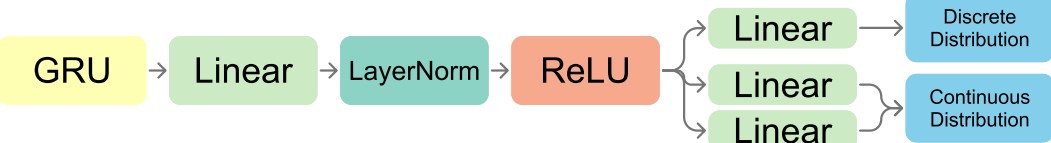

Figure 11: Policy architecture used for training agents in the hybrid action space. The architecture follows Fan et al. (2019), consisting of separate heads for the discrete policy $\pi_{\text{dis}}$ and the continuous policy $\pi_{\text{con}}$, whose outputs are combined to form the joint policy.

We train agents to operate in the hybrid action space by adopting the model architecture proposed by Fan et al. (2019). Given a discrete policy $\pi_{\text{dis}}$ and a continuous policy $\pi_{\text{con}}$, actions are sampled independently from each policy. The joint policy is then expressed as

$$\pi(o, a) = \pi_{\text{dis}}(o, a_{\text{dis}}) \cdot \pi_{\text{con}}(o, a_{\text{con}}),$$

and this factorized probability is used within the learning algorithm. The detailed policy architecture is illustrated in Figure 11.

## C.3 BASELINE HYPERPARAMETERS

Table 6: PPO common hyperparameters

| Hyperparameter | Value |
|---|---|
| Optimizer | Adam (Kingma & Ba, 2014) |
| Anneal Learning Rate | True |
| Batch size | 32 |
| Learning rate | $10^{-4}$ |
| Hidden dim | 256 |
| Hidden layers | 2 |
| Discount factor ($\gamma$) | 0.99 |
| GAE factor ($\lambda$) | 0.95 |
| Activation | ReLU |
| Layer norm (Ba et al., 2016) | True |
| Clip range | 0.2 |
| Update epochs | 5 |
| Max gradient norm | 0.25 |
| Entropy coefficient | 0.1 |

Table 7: PQN common hyperparameters

| Hyperparameter | Value |
|---|---|
| Optimizer | RAdam (Liu et al., 2019) |
| Anneal Learning Rate | True |
| Batch size | 32 |
| Learning rate | $10^{-4}$ |
| Hidden dim | 256 |
| Hidden layers | 2 |
| Discount factor ($\gamma$) | 0.99 |
| $\lambda$ factor | 0.95 |
| Activation | ReLU |
| Layer norm (Ba et al., 2016) | True |
| $\epsilon_{\text{init}}$ | 1.0 |
| $\epsilon_{\text{finish}}$ | 0.05 |
| $\epsilon_{\text{decay}}$ | 0.2 |
| Update epochs | 4 |
| Max gradient norm | 50.0 |
| Reward scale | 100.0 |

Table 8: MAPPO common hyperparameters

| Hyperparameter | Value |
|---|---|
| Optimizer | Adam (Kingma & Ba, 2014) |
| Anneal Learning Rate | True |
| Batch size | 64 |
| Learning rate | $10^{-4}$ |
| Hidden dim | 256 |
| Hidden layers | 2 |
| Discount factor ($\gamma$) | 0.99 |
| GAE factor ($\lambda$) | 0.95 |
| Activation | ReLU |
| Layer norm (Ba et al., 2016) | True |
| Clip range | 0.2 |
| Update epochs | 5 |
| Max gradient norm | 0.25 |
| Entropy coefficient | 0.0 |

Table 9: IPPO common hyperparameters

| Hyperparameter | Value |
|---|---|
| Optimizer | Adam (Kingma & Ba, 2014) |
| Anneal Learning Rate | True |
| Batch size | 64 |
| Learning rate | $10^{-4}$ |
| Hidden dim | 256 |
| Hidden layers | 2 |
| Discount factor ($\gamma$) | 0.99 |
| GAE factor ($\lambda$) | 0.95 |
| Activation | ReLU |
| Layer norm (Ba et al., 2016) | True |
| Clip range | 0.2 |
| Update epochs | 5 |
| Max gradient norm | 0.25 |
| Entropy coefficient | 0.0 |
| Critic coefficient | 0.5 |

## C.4 COMPUTATION TIME

Each experiment was conducted on a single CPU (AMD EPYC 7763, 64 cores) with 512 GB of RAM and a single GPU (NVIDIA RTX 4090, 24 GB memory). We implemented all reinforcement learning algorithms using JAX v0.4.38 and executed them on Debian GNU/Linux 12 (Bookworm).

Table 10: Average wall-clock training time across training methods and algorithm combinations.

| Scenario | Training Method | Algorithm | Training Time |
|---|---|---|---|
| 2F1K2A1H | ST | PPO–MAPPO | 43m |
| | | PPO–IPPO | 39m |
| | | PQN–MAPPO | 43m |
| | | PQN–IPPO | 39m |
| | AT | PPO–MAPPO | 28m |
| | | PPO–IPPO | 26m |
| | | PQN–MAPPO | 28m |
| | | PQN–IPPO | 26m |
| 1K2S | ST | PPO–MAPPO | 40m |
| | | PPO–IPPO | 35m |
| | | PQN–MAPPO | 39m |
| | | PQN–IPPO | 34m |
| | AT | PPO–MAPPO | 23m |
| | | PPO–IPPO | 22m |
| | | PQN–MAPPO | 22m |
| | | PQN–IPPO | 21m |
| 1M2C1P | ST | PPO–MAPPO | 41m |
| | | PPO–IPPO | 36m |
| | | PQN–MAPPO | 39m |
| | | PQN–IPPO | 36m |
| | AT | PPO–MAPPO | 24m |
| | | PPO–IPPO | 23m |
| | | PQN–MAPPO | 24m |
| | | PQN–IPPO | 23m |
| 7F2D1H | ST | PPO–MAPPO | 49m |
| | | PPO–IPPO | 44m |
| | | PQN–MAPPO | 49m |
| | | PQN–IPPO | 44m |
| | AT | PPO–MAPPO | 34m |
| | | PPO–IPPO | 33m |
| | | PQN–MAPPO | 34m |
| | | PQN–IPPO | 33m |

## C.5 CD-PPO DETAIL

CD-PPO is a controlled training setting in which only the TABSUnitComb and TABSUnitDeploy stages are trained with PPO, while the TABSBattleSimulator stage is fixed by a heuristic policy. We define two variants: (i) CD-PPO-Expert, where the heuristic policy is fixed at expert strength from the beginning of training, and (ii) CD-PPO-Schedule, where the heuristic strength is linearly increased until it reaches expert level at the scheduling decay midpoint.

The hyperparameters used for CD-PPO are summarized in Table 11. Performance across all scenarios is reported in Figure 12.

Table 11: Comb&Deploy training common settings

| Setting | Value |
|---|---|
| # Parallel environments | 64 |
| # Max ally units | 10 |
| Total iterations $T$ | 1500 |
| TABSUnitComb rollout steps | 10 |
| TABSUnitDeploy rollout steps | 10 |
| TABSBattleSimulator rollout steps | 512 |
| Initial $\epsilon$ | 0.5 |
| Initial $\epsilon_{\text{ranged}}$ | 1.0 |
| Initial $\epsilon_{\text{healer}}$ | 0.1 |
| Initial $\alpha_{\text{ranged}}$ | 0.5 |
| Initial $\alpha_{\text{healer}}$ | 0.0 |
| Final $\epsilon$ | 0.0 |
| Final $\epsilon_{\text{ranged}}$ | 0.6 |
| Final $\epsilon_{\text{healer}}$ | 0.0 |
| Final $\alpha_{\text{ranged}}$ | 0.0 |
| Final $\alpha_{\text{healer}}$ | 1.0 |
| Enemy $\epsilon$ | 0.1 |
| Enemy $\epsilon_{\text{ranged}}$ | 0.5 |
| Enemy $\epsilon_{\text{healer}}$ | 0.1 |
| Enemy $\alpha_{\text{ranged}}$ | 0.3 |
| Enemy $\alpha_{\text{healer}}$ | 0.85 |
| Scheduling decay midpoint | 0.5 |

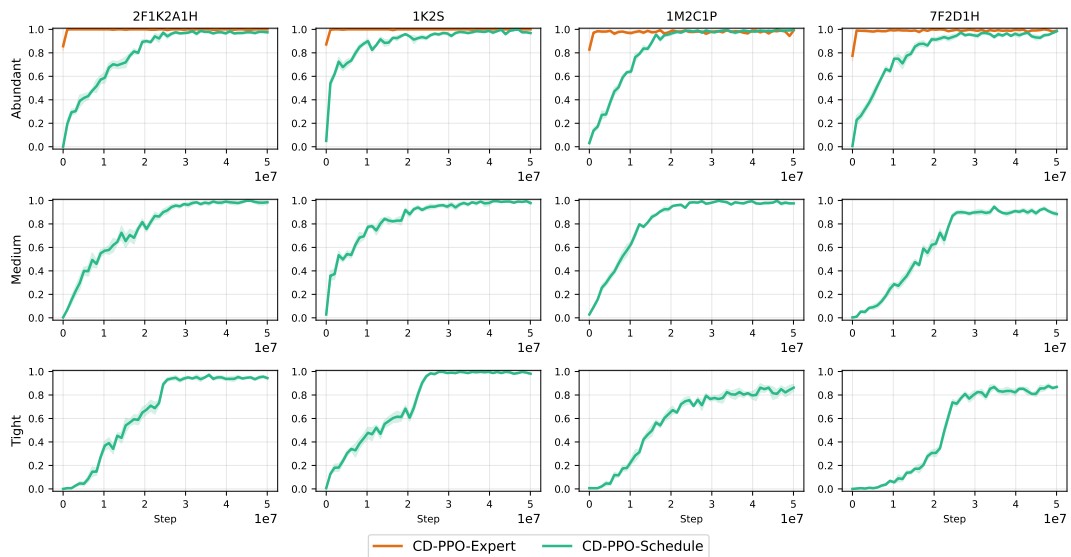

Figure 12: Win rates detail CD-PPO-Expert and CD-PPO-Schedule across varying budget levels. Each point represents the mean episode win rate averaged over 5 random seeds, with shaded regions indicating the standard error.

## C.6 ADDITIONAL RESULTS

We provide several interesting metrics: episode returns (Figure 13), first-kill rate (Figure 14), total episode damage dealt (Figure 15), attack success rate (Figure 16), and average unit counts for each unit (**??**). In all figures, results are averaged over five random seeds, with shaded regions indicating the standard error.

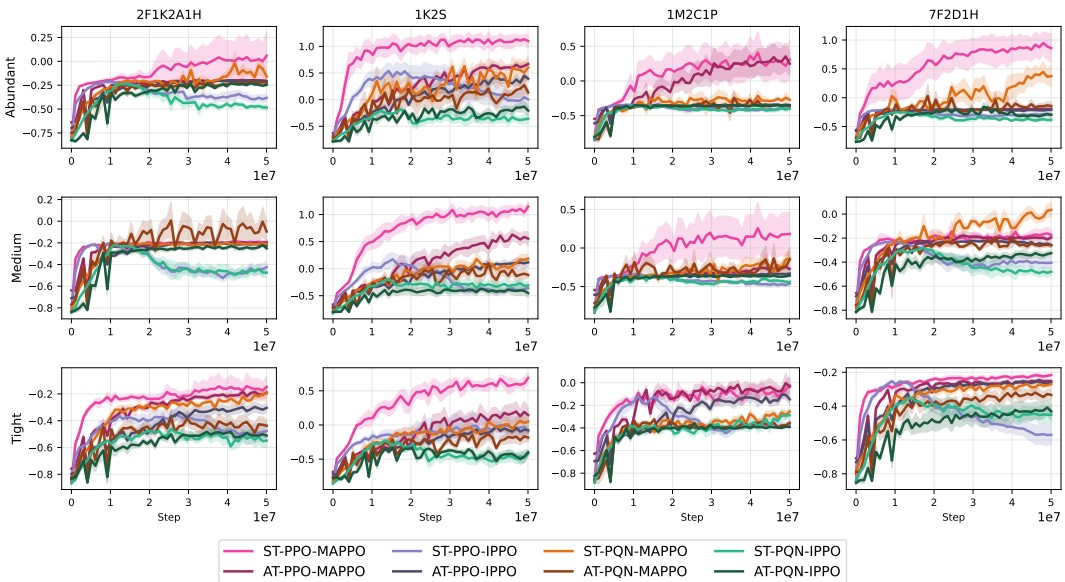

Figure 13: Episode return comparison between alternating training and simultaneous training across varying budget levels.

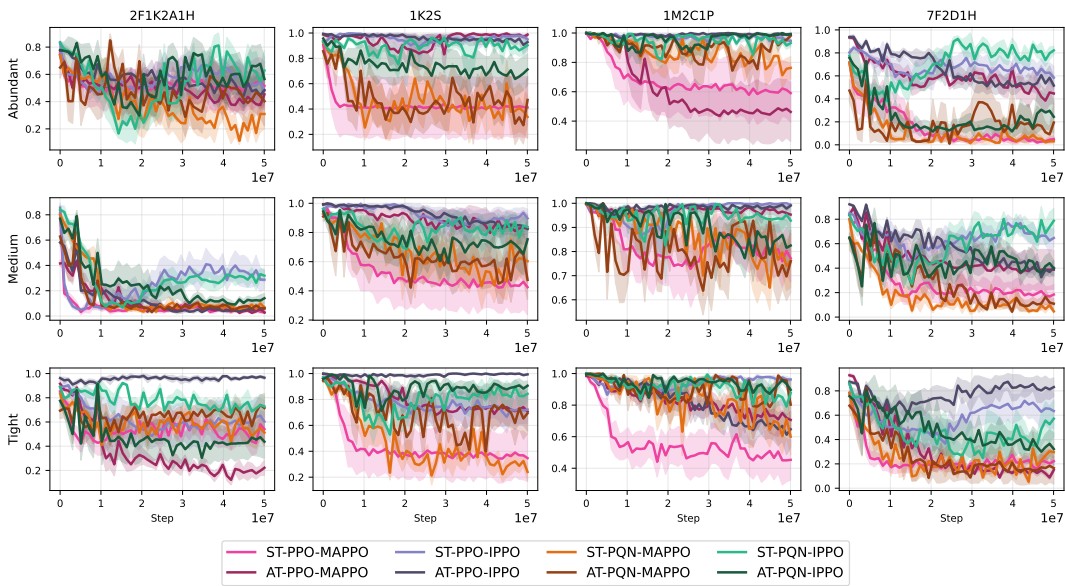

Figure 14: First kill rate comparison between alternating training and simultaneous training across varying budget levels.

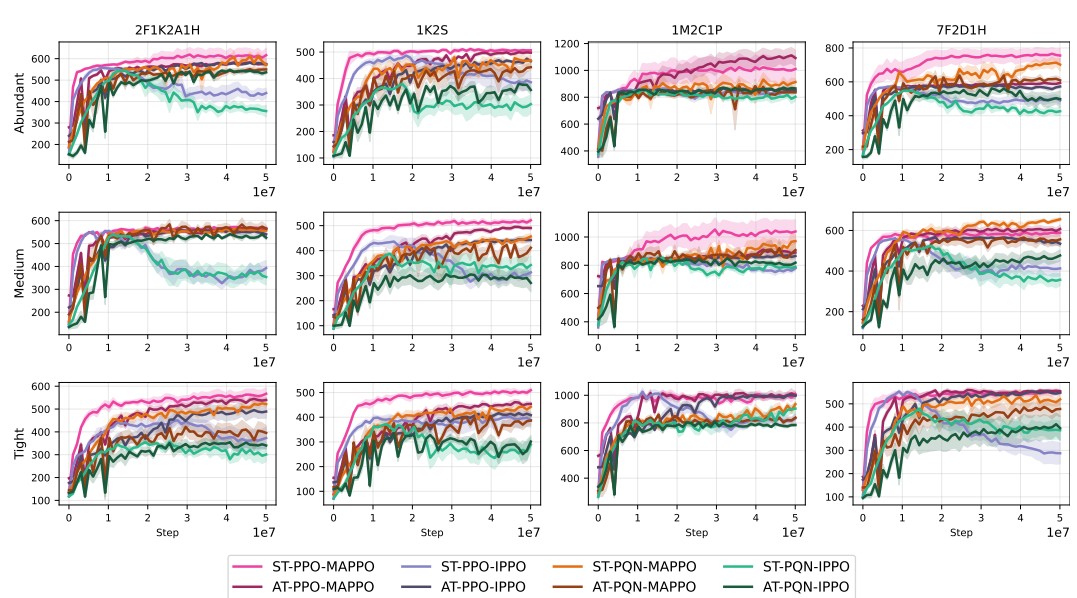

Figure 15: Total episode damage dealt comparison between alternating training and simultaneous training across varying budget levels.

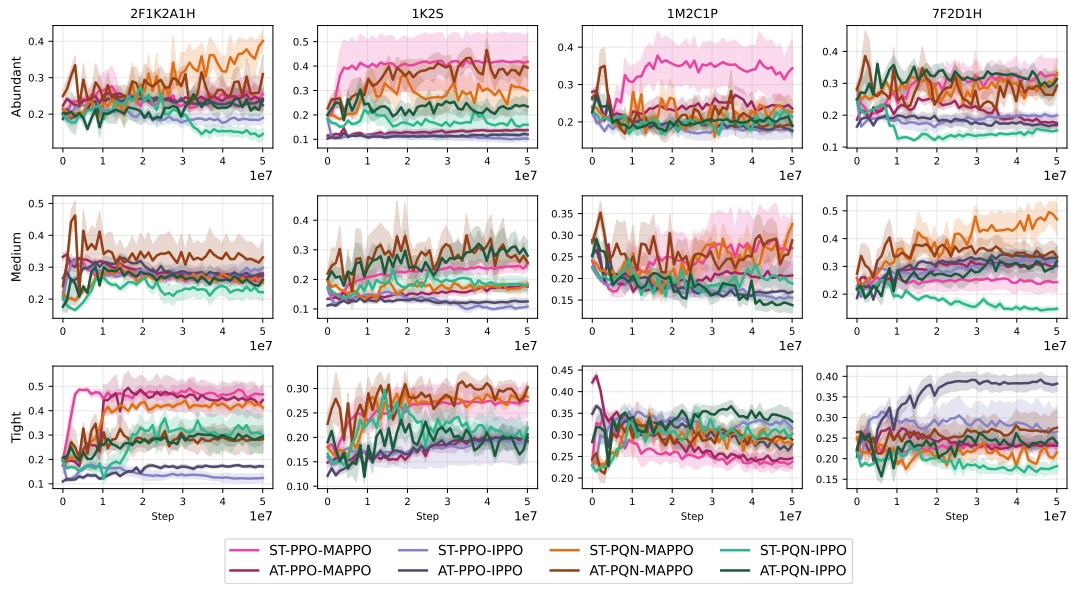

Figure 16: Attack success rate comparison between alternating training and simultaneous training across varying budget levels.

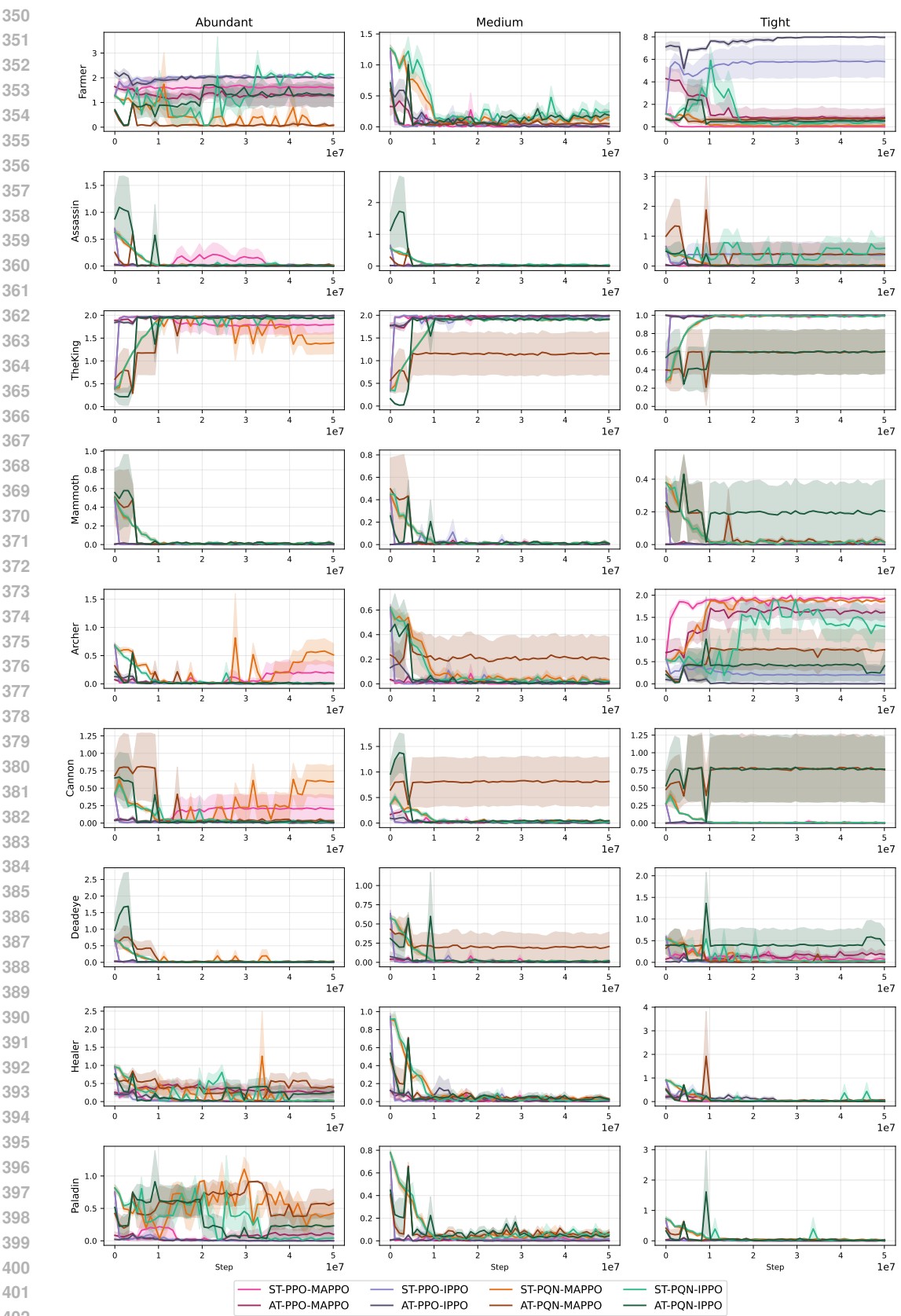

Figure 17: Average usage counts across all units in the 2F1K2A1H scenario

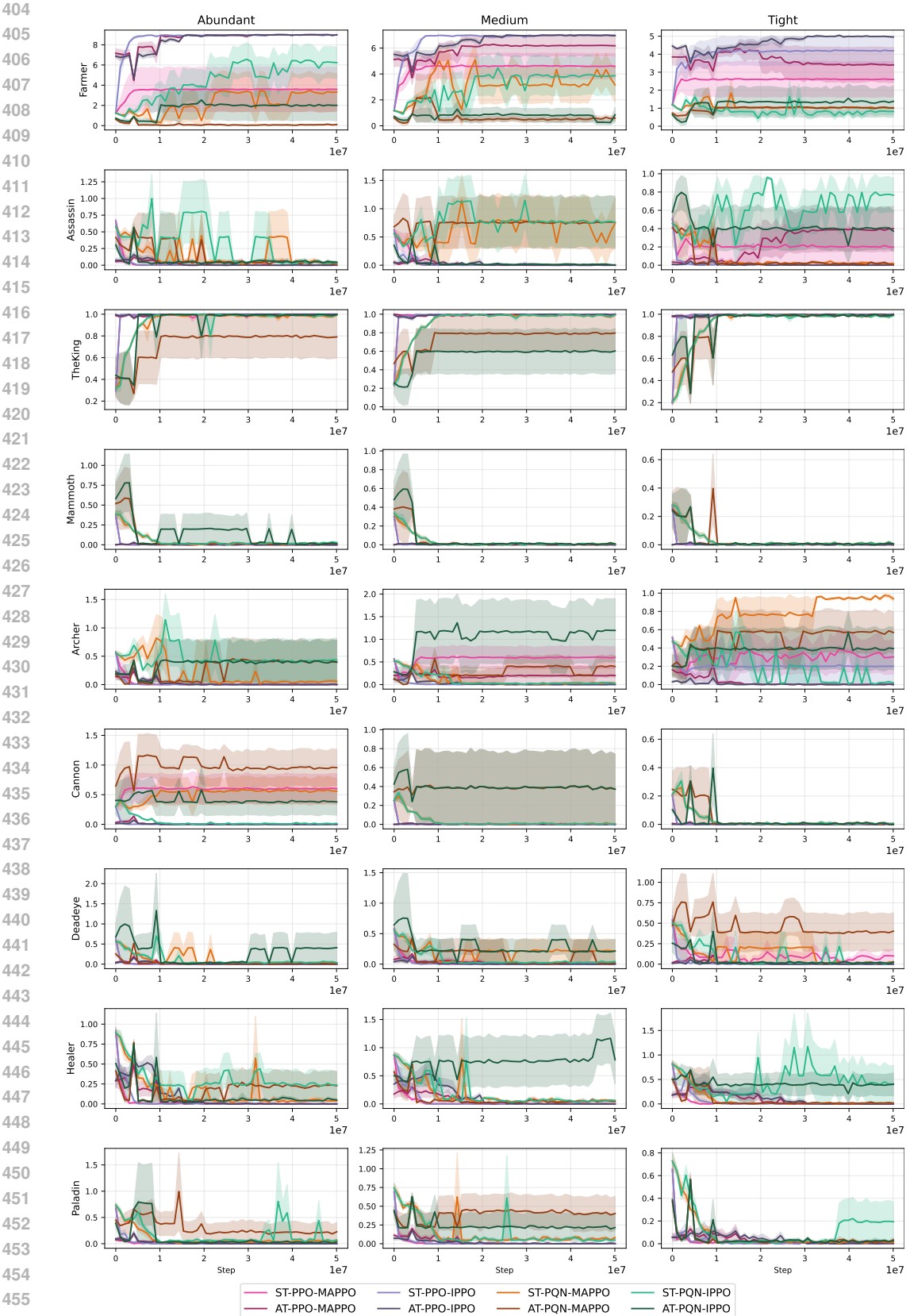

Figure 18: Average usage counts across all units in the 1K2S scenario

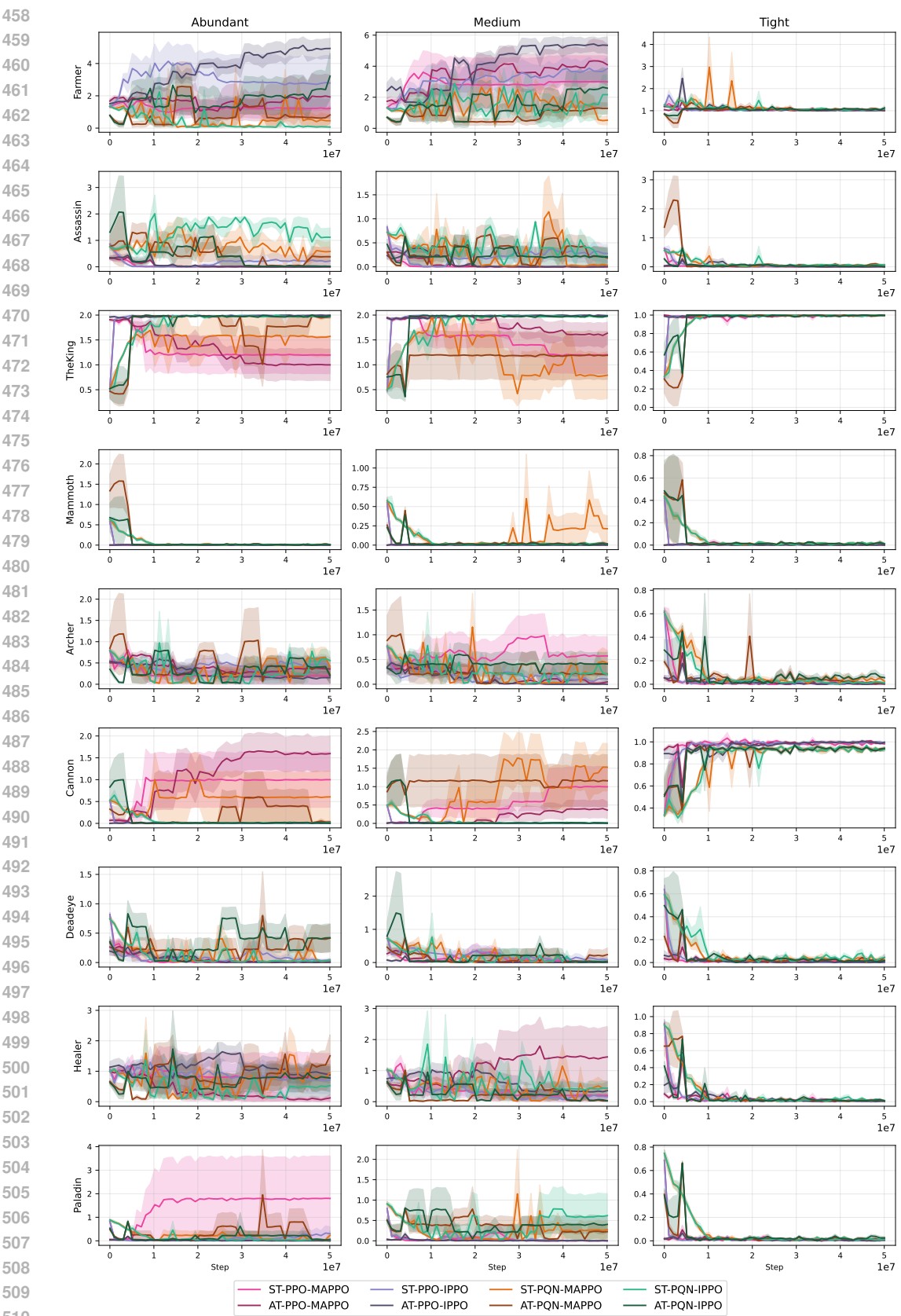

Figure 19: Average usage counts across all units in the 1M2C1P scenario

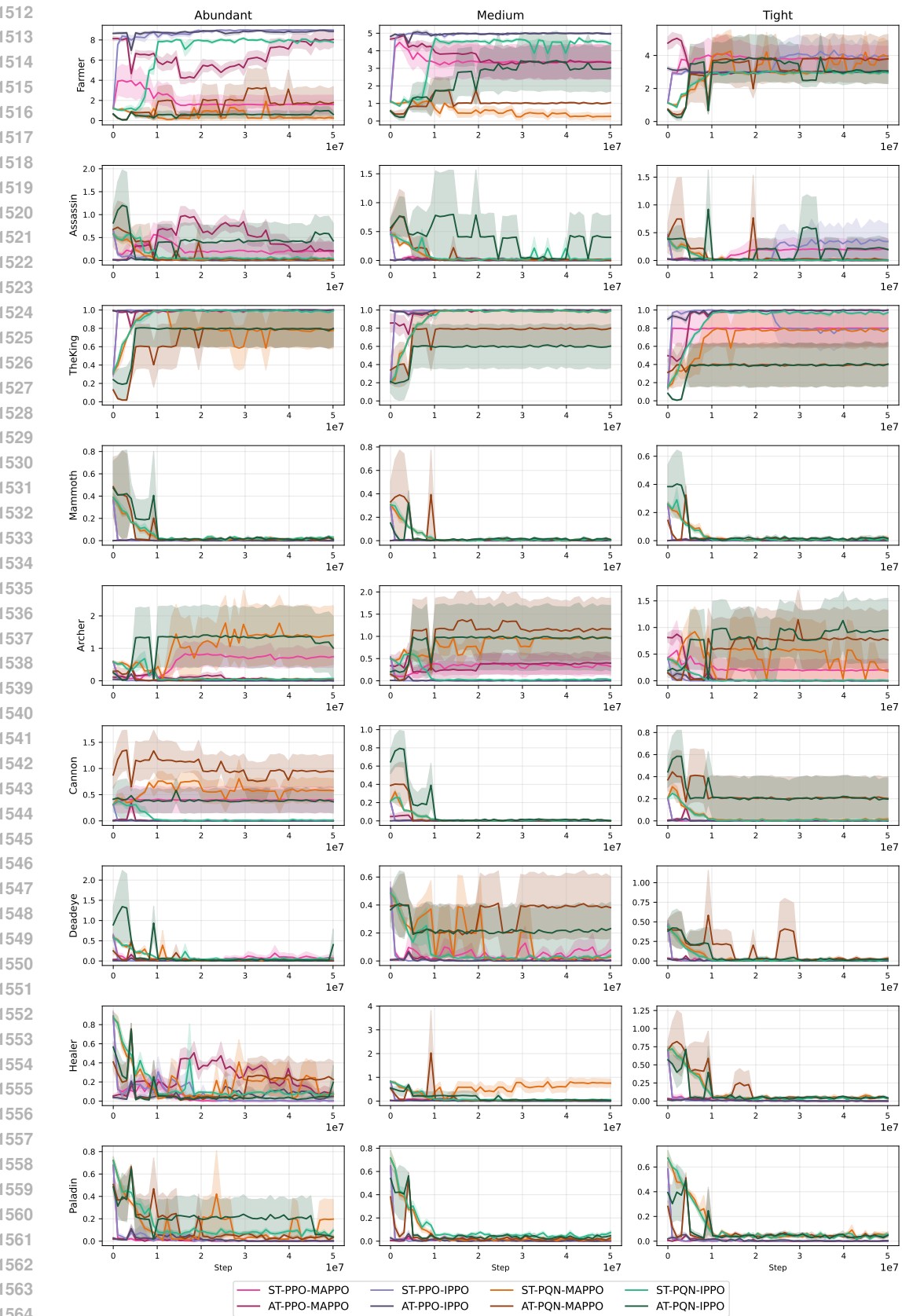

Figure 20: Average usage counts across all units in the 7F2D1H scenario

# D    FULL-SIZED VISUALIZATION

We provide full-sized visualizations of the substages in TABS.

## TABSUnitComb

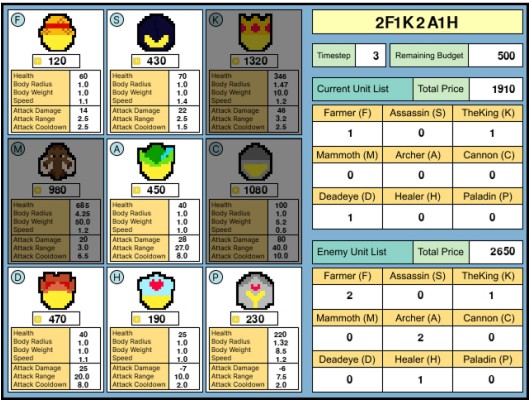

(a) Visualization of TABSUnitComb

## TABSUnitDeploy

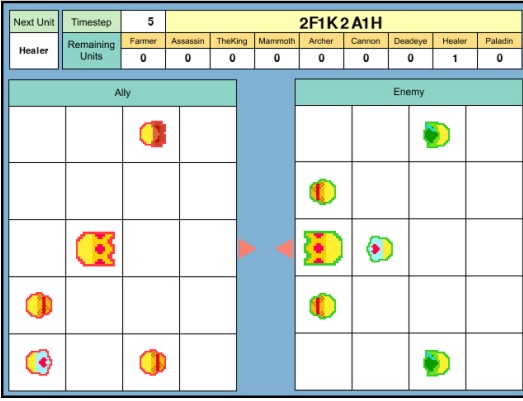

(b) Visualization of TABSUnitDeploy

## TABSBattleSimulator

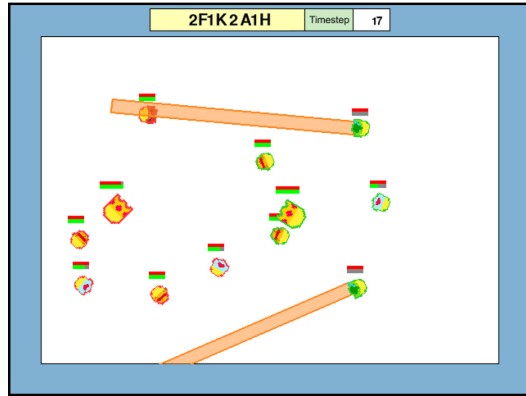

(c) Visualization of TABSBattleSimulator

Figure 21: Full-size Visualization of TABS

