# OpenReview forum: "TABS: Strategic Game-Based Multi-Stage Reinforcement Learning Challenge"
_ICLR.cc/2026/Conference — ICLR 2026 Conference Withdrawn Submission_

### Official Review · Reviewer_ZfE8 · 2025-10-30

**Soundness:** 3
**Presentation:** 3
**Contribution:** 2
**Rating:** 6
**Confidence:** 3

**Summary:**

This paper presents TABS (Totally Accelerated Battle Simulator), a new multi-stage reinforcement learning (RL) benchmark designed to capture sequential dependencies and cross-stage decision-making challenges.
TABS consists of three interlinked stages—unit combination, unit deployment, and battle simulation—where each stage’s outcome determines the input to the next. The environment is implemented fully in JAX, enabling end-to-end GPU acceleration and large-scale parallelization. The authors benchmark several standard RL algorithms (PPO, PQN, MAPPO, IPPO) under two training regimes—simultaneous and alternating—to evaluate performance, stability, and exploration difficulty. Results show that even strong baselines struggle due to entangled credit assignment and exploration bottlenecks, demonstrating the environment’s difficulty and potential as a testbed for hierarchical or cross-stage RL methods.

**Strengths:**

1. Novel multi-stage environment.
TABS explicitly models multi-step interdependencies, addressing an underexplored challenge in current RL benchmarks that typically assume single-stage or loosely connected tasks.
2. High-quality engineering and implementation.
The JAX-based, GPU-accelerated environment is well-structured, scalable, and supports efficient parallel training, showing strong reproducibility and technical completeness.
3. Comprehensive experimental evaluation.
Multiple algorithms, scenarios, and budget settings are benchmarked. The comparisons between simultaneous and alternating training highlight non-trivial learning behaviors.

**Weaknesses:**

1. The benchmark claims strong cross-stage coupling, yet there are no hyperparameters to vary or quantify how much Stage-1/2 choices constrain Stage-3, nor causal/attribution diagnostics linking early decisions to final returns. Scenarios are a small, fixed set(four presets × three budgets(Table 1; Appendix A.4)) rather than a spectrum of coupling strengths, so it’s unclear whether difficulty comes from genuine interdependence or generic MARL hardness. This weakens the central claim that TABS uniquely stresses cross-stage reasoning. It would be beneficial to add cross-stage hyperparameters.
2. For UnitComb/UnitDeploy, the paper assigns the entire final return only to the last action and sets all prior actions’ rewards to 0(Section 4.1 and Algorithms 1–2). This would make it hard to tell whether failures reflect cross-stage credit assignment or just the chosen shaping.
3. All baselines are PPO-family/on-policy (PPO, MAPPO, IPPO, PQN) stitched across stages. Given the paper’s emphasis on long horizons, delayed credit, and exploration, excluding off-policy (e.g., SAC/TD3 variants for hybrid actions), model-based (e.g., Dreamer-style) limits what we can infer about where difficulty truly comes from. As a result, statements like “simultaneous > alternating” or “high variance is inherent” may be method-specific, not environment-intrinsic.
4. The battle opponent is a role-aware heuristic with tunable noise; results fix this heuristic and provide only minimal sensitivity (e.g., “Expert” vs “Schedule” in one scenario) when analyzing exploration/variance. Without a principled sensitivity sweep (or alternate opponent families), robustness and generality remain uncertain.

**Questions:**

1. How sensitive are training outcomes to the stochasticity of the heuristic opponent policy?
2. Is the environment compatible with off-policy or model-based RL pipelines (e.g., JAX-DQN, SAC, TD3, Dreamer)?
3. Could the authors elaborate on how simultaneous vs. alternating updates influence convergence speed or stability?
4. How computationally demanding is training across all scenarios (GPU hours, wall-clock time)?

---

> ### Author Response · Authors · 2025-11-20
>
> We sincerely thank the reviewer for the thorough and constructive feedback.
>
> **Q1. How sensitive are training outcomes to the stochasticity of the heuristic opponent policy?**
>
> We appreciate this insightful comment and plan to run additional experiments to assess how sensitive the training outcomes are to the stochasticity of the heuristic opponent. We vary this stochasticity by adding random movement and rotation or by adjusting attack accuracy. Higher stochasticity is expected to reduce the opponent’s effective attack rate, and confirming this sensitivity will provide useful guidance for researchers who wish to control task difficulty.
>
> **Q2. Is the environment compatible with off-policy or model-based RL pipelines?**
>
> TABS provides the standard transition data and replay support required by off-policy and model-based RL algorithms, such as DQN, SAC, and Dreamer. We plan to run additional experiments with off-policy and model-based baselines to further evaluate this compatibility.
>
> **Q3. Could the authors elaborate on how simultaneous vs. alternating updates influence convergence speed or stability?**
>
> In the alternating update scheme, only one policy is updated at a time while the others remain fixed. This produces more stable learning dynamics because the target changes slowly, although it can increase the chance of converging to suboptimal solutions. In contrast, simultaneous updates adjust all policies together, which generally leads to faster improvement in practice. As shown in Figure 6, the simultaneous methods tend to exhibit quicker rises in win rate, although this joint optimization can make the learning curves somewhat less smooth.
>
> **Q4. How computationally demanding is training across all scenarios (GPU hours, wall-clock time)?**
>
> All experiments were conducted on a single RTX 4090 GPU. The computational cost for each scenario is reported in Appendix C.4, and Figure 8 shows how the cost scales with the number of agents.
>
> **Q5. Assigning the entire final return only to the last action would make it difficult to determine whether failures stem from multi-stage credit assignment issues or simply from the chosen shaping.**
>
> Due to the characteristics of our environment, the final reward is available only after all stages have been completed. Assigning this outcome to each independent stage policy is therefore a natural design choice. Consequently, the difficulty in credit assignment is not an artifact of arbitrary reward shaping, but rather an inherent challenge arising from the environment’s structure.

---

> > ### Comment · Reviewer_ZfE8 · 2025-11-23
> >
> > Thank you for you detailed response. Figure 8 should be showing the speed, not the cost. I'll maintain my score since there is not additional experiment added.

---

### Official Review · Reviewer_xY4X · 2025-10-30

**Soundness:** 2
**Presentation:** 1
**Contribution:** 1
**Rating:** 2
**Confidence:** 4

**Summary:**

The paper introduces TABS (Totally Accelerated Battle Simulator), a multi-stage reinforcement learning environment inspired by strategic battle games. The environment consists of three sequential stages: unit selection, unit deployment, and real-time battle simulation. The authors implement the environment in JAX to enable GPU-accelerated training and evaluate several baseline algorithms (PPO, PQN with MAPPO/IPPO) using both simultaneous and alternating training methods.

**Strengths:**

- Timely engineering contribution: The JAX implementation enabling GPU-accelerated environment simulation addresses a practical bottleneck in RL research, achieving significant speedup as demonstrated in Figure 8.
- Comprehensive experimental analysis: Detailed investigation across multiple algorithms (Section 4.2) with both simultaneous and alternating training methods, providing insights into the challenges of multi-stage learning.
- Thoughtful reward design (Line 199): The shared reward proportional to health ratio differences and binary win-loss outcome is well-motivated for the battle simulation stage.
- Clear visualizations: Section 3.1 provides helpful visual representations of the environment stages, particularly the field-of-view constraints and their impact on initial observations.
- Role-appropriate heuristic policy: The differentiated behaviors for different unit types (Section 3.3) provides more realistic and challenging opponent strategies than uniform policies.

**Weaknesses:**

- Unclear abstract and motivation: The abstract fails to clearly define what constitutes a "complex multi-stage environment" or why existing benchmarks are insufficient. The phrase "tightly coupled decision points" is vague and never formally defined.
- Missing formal definitions: The main paper lacks mathematical formulation of what constitutes a "stage" and how stages differ from standard MDPs with varying state spaces. The interdependencies between stages are described informally without clear specification of the coupling mechanisms.
- Limited novelty in multi-stage concept: The paper presents multi-stage decision-making as contribution with limited comparisons and discussion of prior works.
- Insufficient RL background: The paper focuses heavily on game mechanics and engineering details while providing minimal background on relevant MARL concepts. Related work is relegated to Appendix B (Line 127), which should be in the main text.
- How are policies structured to handle different observation/action spaces across stages (Lines 134, 351)?
- What exactly is a "stage-conditioned mixture of stage policies" (Line 363)?
- How is an episode defined relative to the "pipeline" (Line 365)?
- Limited scientific contribution: The paper reads more as an environment release than a research contribution. The experiments primarily demonstrate that existing algorithms struggle with the environment but provide limited insights into why or how to address these challenges.
- Exploration analysis lacks depth: While Section 4.3 identifies exploration challenges, the analysis relies only on basic entropy regularization without investigating modern exploration methods (RND, disagreement, curiosity-driven approaches).

**Questions:**

- Line 050: "Aligned and integrated" with respect to what? This metaphor needs clarification or removal.
- Line 055: Can you provide a formal definition of "stage" early in the introduction? How does this differ from standard episodic RL with varying state spaces?
- Line 068: How is interdependence between stages different from any sequential decision problem? The chess tournament analogy (winning individual games to win tournament) suggests this isn't unique.
- Line 134: When stages have different observation/action spaces, are they padded to equal dimensionality? How does the policy architecture handle these variations?
- Line 135: Is "entire pipeline" synonymous with completing all stages sequentially? Please define this term clearly.
- Line 351: What does "distinct" mean formally in terms of observation/action spaces? Are they different types, dimensions, or both?
- Line 363: Please provide the mathematical formulation for "stage-conditioned mixture of stage policies".
- Line 365: How is an episode defined? Does it span all stages or is each stage a separate episode?
- Line 412: Have you conducted ablations on entropy regularization? Have you tested modern exploration methods (RND, disagreement-based, curiosity-driven) given that entropy-based exploration has known limitations?

---

> ### Author Response · Authors · 2025-11-20
>
> We sincerely appreciate your thorough and detailed review. We hope the following responses will address your concerns and provide the clarification you seek.
>
> **Q1. How is the stage-conditioned mixture of policies implemented? In particular, how are the policies structured to accommodate the different observation and action spaces across stages?**
>
> We apologize that some formal definitions for these components were missing in the main text. We will move the key definitions from Appendix A into the main text for clarity. The stage-conditioned mixture is implemented by using separate policy networks for each stage, each equipped with its own architecture. This allows the framework to handle the differing observation and action spaces across stages, as each policy network is specialized for the requirements of its corresponding stage.
>
> **Q2. Missing Related works**
>
> Existing multi-stage environments (e.g., Procgen, MineDojo) typically rely on a homogeneous observation–action space, and therefore do not require fundamentally different agents or policies across stages. In contrast, our environment is characterized by heterogeneous observation and action spaces across stages, necessitating distinct policy architectures and making the overall problem significantly more challenging than prior multi-stage settings. We will expand the Related Works section to more thoroughly analyze prior environments and clearly articulate how our setting differs from them.
>
> **Q3. The experiments primarily demonstrate that existing algorithms struggle with the environment but provide limited insights into why or how to address these challenges.**
>
> We appreciate this comment. Our current experiments focus on showing that standard algorithms face clear difficulties in this environment, but we agree that the present analysis provides limited insight into the underlying causes or possible remedies. A more systematic examination of the environment’s unique challenges and their influence on learning would be valuable. Future work is expected to explore these aspects in greater depth and incorporate diagnostics that clarify why these difficulties arise and how they might be mitigated.
>
> **Q4. The analysis relies only on basic entropy regularization without investigating modern exploration methods.**
>
> We appreciate this comment. As you noted, basic entropy regularization has clear limitations in our setting. We will include baselines that incorporate a range of modern exploration methods to further examine their impact on performance.
>
> **Q5. How is interdependence between stages different from any sequential decision problem?**
>
> TABS consists of three separate stages with distinct properties, including different state and action spaces as well as single- and multi-agent settings. In this setting, using separate policies for each stage is reasonable. In previous multi-stage environments such as Procgen, training stage-specific policies and then combining them can more easily yield a fully optimal policy, but this is not the case in TABS.
>
> **Q6. Writing Concern**
>
> We appreciate your feedback regarding the unclear parts of our writing. We will revise the terminology and phrasing to ensure that the intended meaning is conveyed more clearly throughout the paper.

---

> > ### Comment · Reviewer_xY4X · 2025-11-25
> > **Many concerns not addressed appropriately**
> >
> > I thank the authors for their response. However, I find that most of my concerns remain unaddressed:
> >
> > **Regarding Q3 (limited insights into algorithmic challenges):** The authors acknowledge that existing algorithms struggle but provide no concrete analysis of *why* this occurs or *how* future methods might address these challenges. Venues like ICLR have a history of publishing benchmark papers that include not only challenging environments but also initial algorithmic insights or "signs of life" that point toward solutions. Without such contributions, the paper remains primarily an engineering artifact rather than a research contribution.
> >
> > **Regarding Q4 (exploration methods):** The authors' response does not clarify *what* modern exploration methods will be tested, *when* or *how* they will be incorporated, or whether this will happen during the discussion period. This shortcoming could potentially be resolved, but no concrete commitment has been made.
> >
> > **Critically, none of my questions regarding formal definitions were addressed:**
> >
> > - The mathematical formulation of "stage" and its distinction from standard MDPs with varying state spaces remains absent
> > - The formal definition of "stage-conditioned mixture of stage policies" (Line 363) was not provided
> > - How the policy architecture handles heterogeneous observation/action spaces across stages (Lines 134, 351) remains unclear: Are these implemented as separate subnetworks that get activated for each stage? How does the switching mechanism work?
> > - The episode definition relative to the "pipeline" (Lines 135, 365) was not clarified
> > - The vague terminology ("aligned and integrated," "tightly coupled," "distinct") remains undefined
> >
> > These formalizations are standard in RL literature and essential for reproducibility and theoretical understanding.
> >
> > Given the modest responses to specific technical questions and the complete lack of engagement with the formalization concerns central to my review, I maintain my score until these issues are substantively addressed. Additionally, the authors did not make use of the option to update the PDF to demonstrate how these concerns have been addressed.

---

### Official Review · Reviewer_qH3m · 2025-10-30

**Soundness:** 2
**Presentation:** 3
**Contribution:** 2
**Rating:** 4
**Confidence:** 3

**Summary:**

TABS is a strategy-game-inspired, multi-stage, accelerated RL environment written in JAX. It provides configurable environments and four default scenarios (configurations). The first two stages are combinatorial in nature, whereas the final stage can be cast as a multi-agent problem. TABS presents three challenges: structured exploration, heterogeneous/mixed action spaces, and delayed rewards.

**Strengths:**

I think TABS can contribute to both single- and multi-agent RL research in the following ways:
- RL methods focusing on structured exploration and long-term credit assignment can leverage its core challenges.
- Heterogeneous action spaces and multi-stage environments are under explored. The multi-stage aspect of the environment led the authors to explore an alternative training paradigm, i.e., alternating training.
- A JAX-friendly and configurable environment that can be integrated to pure JAX RL implementations. Plus, TABS provides built-in visualizations.

**Weaknesses:**

My main concern is how TABS differentiates itself within the vast space of accelerated RL environments.

- Although I am unsure how best to measure it, I would like to see quantitative comparisons to existing environments such as Atari, Jumanji [3] (the first two stages share a combinatorial flavor), and the MuJoCo Control Benchmarks, particularly in exploration and credit assignment. See [1] for a similar analysis for agents.
- I was expecting a gym(-nax [2])-like API for making environments. The current setting leaves the environment configurable, but for standardization and broader adoption, the authors may consider a list of fixed/predefined configurations with accompanying names and use a `tabs.make("<scenario-name>")`-like API for initialization.
- Although the source is easy to navigate, I would like to see more detailed documentation in README regarding the stages, environment creation, observation and action spaces, and reward computation.

Also see some of the questions.

[[1](https://openreview.net/pdf?id=rygf-kSYwH)] Osband, Ian, et al. "Behaviour suite for reinforcement learning." arXiv preprint arXiv:1908.03568 (2019).

[[2](https://github.com/RobertTLange/gymnax)] Lange, Robert Tjarko. Gymnax: A JAX-Based Reinforcement Learning Environment Library. 0.0.4, 2022, github.com/RobertTLange/gymnax.

[[3](https://github.com/instadeepai/jumanji)] Bonnet, Clément, et al. "Jumanji: a diverse suite of scalable reinforcement learning environments in jax." arXiv preprint arXiv:2306.09884 (2023).

**Questions:**

- I would like to see a discussion of how TABS can contribute to the development of new RL algorithms. How much of TABS’s difficulty stems from its multi-stage nature? How impactful are early-stage decisions on the BattleSimulator stage, and how could this be measured?
- Do you provide starter opponents and configurations with **increasing** difficulty (e.g., few available units and positions) so practitioners can quickly test approaches and gradually scale up?
- Can TABS be used to measure generalization (or easily configured to do so), for example via procedurally generated configurations?
- Is there an end-to-end scenario that unifies the three action and observation spaces into a single three-stage `env` in TABS? If not, what are the main challenges in providing such an environment?
- How fast is TABS compared to other JAX-based accelerated environments, e.g., in steps/second, frames/second, and episodes/second?
- Did you apply any tuning process (manual or otherwise) to balance unit strength and price, for example, to avoid trivial solutions where one unit type consistently dominates?
- (small note) Is `run_example.py` under the `tabs` directory using a depreciated object `default_tabs_conf`  from `tabs.scenarios`? I could not run it.

---

> ### Author Response · Authors · 2025-11-20
>
> We thank the reviewer for the thorough and constructive comments. We hope we can address your concerns below.
>
> **Q1. How impactful are early-stage decisions on the BattleSimulator stage, and how could this be measured?**
>
> In Section 4.3, we discuss the impact of the early stages by comparing CD-PPO-Expert and CD-PPO-Schedule. These results show that decisions made in the early stages are influenced by battle expertise. We also provide the average unit counts across all units in Appendix C.6. However, we appreciate your point that a more appropriate measurement is needed to better highlight this challenge, and we will take this into consideration.
>
> **Q2. Do you provide starter opponents and configurations with increasing difficulty?**
>
> By adjusting the action mask provided in UnitComb and UnitDeploy, users can gradually restrict the types of units that can be purchased or limit the allowable deployment regions, thereby creating configurations with increasing difficulty. In addition, as demonstrated in Section 4.3, we can control the expertise level of a heuristic policy. We will extend this functionality to opponent agents as well, allowing users to flexibly adjust difficulty levels through configurable options.
>
> **Q3. Can TABS be used to measure generalization (or easily configured to do so)?**
>
> TABS can be readily adapted to construct diverse settings suitable for studying generalization, including procedurally generated scenarios enabled by configurable parameters such as unit specifications and battlefield size. We will add a description of this capability to the appendix.
>
> **Q4. Is there an end-to-end scenario that unifies the three action and observation spaces into a single three-stage in TABS?**
>
> We implemented TABS by separating Python classes for clarity and modularity, as each stage has a distinct action and observation space. However, since our environments are designed to be easily modifiable, it is indeed possible to construct a unified environment class that integrates all three stages into a single end-to-end scenario.
>
> **Q5. How fast is TABS compared to other JAX-based accelerated environments, e.g., in steps/second, frames/second, and episodes/second?**
>
> It is not straightforward to compare TABS directly with other JAX-based accelerated environments because the setting differs in several ways. However, we will provide throughput metrics so that users can still gauge the relative speed of the environment.
>
> **Q6. Did you apply any tuning process (manual or otherwise) to balance unit strength and price, for example, to avoid trivial solutions where one unit type consistently dominates?**
>
> We manually tuned the configuration by estimating the win rate of the heuristic policy and adjusting unit strengths and prices to target roughly a 50:50 win rate when both teams have the same total cost. We agree that a more systematic balancing procedure would be valuable for reducing trivial dominance between unit types. We expect to explore such approaches in future work.
>
> **Q7. TABS code implementation**
>
> We apologize for the inconvenience. We will update the script and configuration to ensure everything runs properly, and we will address this issue as soon as possible. We will also update the README promptly.

---

### Official Review · Reviewer_9qk3 · 2025-11-01

**Soundness:** 2
**Presentation:** 2
**Contribution:** 2
**Rating:** 2
**Confidence:** 2

**Summary:**

The authors describe TABS a multi-stage RL challenge.
The environment is based on a battle simulator environment with multi-distinct stages that are dependent on one another
Stages are difference where it both single and multi-agent problems
The environment's goal is to test the limitations of standard RL algorithms, which struggle to coordinate effectively across stages, and provides a benchmark for testing multi-stage decision-making.

**Strengths:**

* The environment described an interesting multi-stage long-horizon RL problem
* The environment is has distinct but interdependent stages which is not similar to existing environments

**Weaknesses:**

* The authors focus a lot on describing the details of the env, including how unit properties and how they interact (section 3.2 and 3.2), but doesn't really focus on the insights from the multi-stage learning problem.
* Comprehending the details of the environment was difficult for me as I am not familiar with Landfall games
* The authors provide baseline results but doesn't reveal much insights from the experiments. It is difficult to understand how to performance of different stages affects the final performance, the effects of simultaneous vs alternating training

**Questions:**

* Were other RL algorithms explored?
* What new insights about multi-stage RL are revealed by this environment that cannot be studied in existing benchmarks?
* What are the main advantages of using TABS compared to say adding Units comb and Unit deploy stages in the SMAC / SMAX?

---

> ### Author Response · Authors · 2025-11-20
>
> Thank you for your thoughtful comments. Please find the responses to your questions below.
>
> **Q1. Lack of insights into the multi-stage learning problem**
>
> We appreciate your observation that our insights into the multi-stage problem are currently limited. As we discuss in the paper, many real-world environments consist of several distinct stages, such as cooking or playing sports. Moreover, unlike prior works (e.g., Procgen, MineDojo), each stage in TABS is heterogeneous, involving different state/action spaces and both single- and multi-agent settings. We will improve our analysis and expand our insights in future revisions.
>
> **Q2. Other RL algorithms**
>
> We thank you for your constructive feedback. We will conduct additional experiments with off-policy and model-based baselines to further validate this compatibility.
>
> **Q3. What are the main advantages of using TABS compared to say adding UnitComb and UnitDeploy stages in the SMAC/SMAX?**
>
> In TABS, the results of each stage are passed as an argument to the initial state distribution of the next stage. When we attach the first two stages to SMAC/SMAX, they take a form similar to TABS. However, initial states play a more critical role in TABS due to its partially fan-shaped observations, as discussed in Section 3.1 and Appendix A.1. Moreover, unit composition is influential because we provide a role-appropriate heuristic policy for the battle stage. Therefore, TABS emphasizes the interdependency between stages.
>
> **Q4. TABS Landfall games**
>
> We apologize for not clearly explaining the Landfall TABS game that inspired our work. Landfall Games’ TABS is a battle simulation game in which players combine and deploy units with different specifications and unique attack skills within a limited budget to defeat their opponents. We will include more detailed information about the game in the Appendix.

---

### Author Response · Authors · 2025-11-26
**Global Comment for reviewers**

We sincerely appreciate all reviewers for their constructive and detailed comments.
We recognize that our paper requires significant revision to improve its clarity and scientific contribution. Therefore, we have decided to withdraw the manuscript for future resubmission.
Thank you again for your time and effort in reviewing our work.

---

### Note · Authors · 2025-11-26

I have read and agree with the venue's withdrawal policy on behalf of myself and my co-authors.